# Short- and long-range *cis* interactions between integrated HPV genomes and cellular chromatin dysregulate host gene expression in early cervical carcinogenesis

Ian J. Groves[1,¤a]*, Emma L. A. Drane[1], Marco Michalski[2], Jack M. Monahan[3], Cinzia G. Scarpini[1], Stephen P. Smith[1], Giovanni Bussotti[3,¤b], Csilla Várnai[2,¤c], Stefan Schoenfelder[2], Peter Fraser[2,4], Anton J. Enright[1,3], Nicholas Coleman[1]

**1** Department of Pathology, University of Cambridge, Cambridge, United Kingdom, **2** Nuclear Dynamics Programme, Babraham Institute, Cambridge, United Kingdom, **3** EMBL-European Bioinformatics Institute, Wellcome Trust Genome Campus, Hinxton, United Kingdom, **4** Department of Biological Science, Florida State University, Tallahassee, Florida, United States of America

☯ These authors contributed equally to this work.
¤a Current address: Cambridge Institute of Therapeutic Immunology and Infectious Disease and Department of Medicine, University of Cambridge School of Clinical Medicine, Cambridge Biomedical Campus, Cambridge, United Kingdom
¤b Current address: Institut Pasteur, Bioinformatics and Biostatistics Hub, Paris, France
¤c Current address: Centre for Computational Biology, University of Birmingham, Birmingham, United Kingdom
* ijg25@cam.ac.uk

**Data Availability Statement:** All data supporting the findings of this study are present within the article and its Supplementary Information files,

## Abstract

Development of cervical cancer is directly associated with integration of human papillomavirus (HPV) genomes into host chromosomes and subsequent modulation of HPV oncogene expression, which correlates with multi-layered epigenetic changes at the integrated HPV genomes. However, the process of integration itself and dysregulation of host gene expression at sites of integration in our model of HPV16 integrant clone natural selection has remained enigmatic. We now show, using a state-of-the-art 'HPV integrated site capture' (HISC) technique, that integration likely occurs through microhomology-mediated repair (MHMR) mechanisms via either a direct process, resulting in host sequence deletion (in our case, partially homozygously) or via a 'looping' mechanism by which flanking host regions become amplified. Furthermore, using our 'HPV16-specific Region Capture Hi-C' technique, we have determined that chromatin interactions between the integrated virus genome and host chromosomes, both at short- (<500 kbp) and long-range (>500 kbp), appear to drive local host gene dysregulation through the disruption of host:host interactions within (but not exceeding) host structures known as topologically associating domains (TADs). This mechanism of HPV-induced host gene expression modulation indicates that integration of virus genomes near to or within a 'cancer-causing gene' is not essential to influence their expression and that these modifications to genome interactions could have a major role in selection of HPV integrants at the early stage of cervical neoplastic progression.

with all sequencing data deposited in the ArrayExpress database at EMBL-EBI (www.ebi.ac.uk/arrayexpress) under accession numbers: E-MTAB-10152; E-MTAB-10154; E-MTAB-10155.

**Funding:** This work was supported by Cancer Research UK (www.cancerresearchuk.org) Programme Award (A13080) to NC. ELAD was supported by a PhD studentship from The Pathological Society of GB & NI (www.pathsoc.org) awarded to IJG and NC. The funders had no role in study design, data collection and analysis, decision to publish, or preparation of the manuscript.

**Competing interests:** The authors have declared that no competing interests exist.

## Author summary

The integration of human papillomaviruses (HPVs) into host chromosomes is a major feature of HPV-associated cancers, however the process by which this occurs and subsequently drives carcinogenesis is incompletely understood. Here, we devised a state-of-the-art HPV16 genome-specific DNA capture technology to precisely determine the host integration sites at a nucleotide resolution, such that we confirm the mechanism of microhomology-mediated repair (MHMR) during both 'direct' and 'looping' integration of HPV16 genomes into the host. Furthermore, our technology detects both short- and long-range interactions between HPV16 and host chromatin after virus integration, which correlates with dysregulation of host gene expression at distances up to 500kbp from the integration site. This means that HPV16 genomes can directly affect host gene expression much further away on host chromosomes than initially thought, which may lead to competitive growth advantages for certain integrated clones. Therefore, our study provides further insight into the mechanisms by which papillomaviruses are able to initiate and drive cervical carcinogenesis at an early stage after HPV integration.

## Introduction

Human papillomavirus (HPV) infection is associated with the development of around 5% of all human cancers, with ~690,000 cases arising annually worldwide[1]. Of these, ~570,000 are cancers of the cervix, which usually present as squamous cell carcinomas (SCCs), developing through clonal selection of cells with a competitive growth advantage from precursor squamous intraepithelial lesions (SILs) and ultimately leading to ~260,000 deaths globally[2–5]. The association of high-risk HPV (HRHPV) infections with cervical SCCs is over 99.9% and, as such, makes HPV the etiological agent associated with cervical carcinomas[6,7]. The treatment of HPV-associated carcinomas has changed little over the past 30 years and, despite current vaccination programs against HPV, new therapies are necessary for an aging unvaccinated population.

The genome of HPV usually exists as an extra-chromosomal episome of around 7.9 kilobases (kb) at a copy number of around 100 per cell in squamous epithelial lesions as part of the normal virus lifecycle[8,9]. Although development of cervical SCC with an episomal HPV genome can occur[10], progression of disease towards cervical carcinoma is more often associated with integration of the fractured, double stranded DNA (dsDNA) HRHPV genome into that of the host, occurring in around 85% of cases[11–13]. The integration process usually involves the disruption of the HPV *E2* gene and, with loss of this trans-repressive protein product, leads to dysregulation of virus gene expression from the early promoter[5,11,14]. Despite this process usually resulting in an increase in HPV oncogene expression associated with cervical SCCs, our previous work has shown that the genomes at initial integration events prior to selection can have levels of expression similar to, or lower than, parental episomal cell lines [15]. Indeed, this is also reflected in the growth rates of these cloned cell lines, which also span that of the parental episomal cell line such that an apparent competitive growth advantage is not always evident[15].

The control of HPV gene expression in the productive lifecycle of the virus occurs in a differentiation-dependent manner associated with the position of the virus within the infected stratified epithelium[5]. The necessary expression of the HPV oncogenes *E6* and *E7* in the basal layer cells occurs through transcriptional initiation at the virus early promoter (p97 in

the case of HPV16). This is controlled by the interaction of various host transcription factors with regulatory elements within the long control region (LCR) and modification to the local chromatin structure[5,16,17]. The binding of these factors is known to become modified as infected cells differentiate such that the late promoter (p670 for HPV16) is stimulated and late virus gene expression ensues[18,19]. In concert, changes to chromatin structure are known to occur as late HPV gene expression becomes activated, driven through the modification of histone post-translational modifications (PTMs) at HPV genome-associated nucleosomes[18,20]. These histone PTMs have also been found associated with the enhancer and promoters of HPV16 episomes during progression of *in vitro* neoplastic progression, with acetylation of both histone H3 and H4 (H3ac and H4ac, respectively) accumulating during the initial stages of phenotypic progression to SCC[10]. Work from our lab investigating the integrated HPV16 genome has previously shown differential association of histone modifications with the virus LCR and early genes corresponding to levels of virus transcript per template[15]. Subsequent studies have shown that multi-layered epigenetic modifications to the integrated HPV genome are associated with the recruitment and activation of RNA polymerase II (RNAPII), thereby determining the level to which HPV oncoprotein expression occurs[21]. These modifications include levels of DNA methylation, further histone PTMs and associated enzymes, as well as nucleosome positioning and the presence of chromatin remodelling enzymes and transcriptional activators, such as the P-TEFb complex, directly at the integrated virus templates[5,21].

However, as implied, cervical SCC is not always associated with high virus oncogene levels and may be driven independent of this factor, for example through host gene changes[22]. HPV appears to integrate into certain sites across the human genome more often than others, so called 'integration hotspots', associated at times with chromosome fragile sites (CFSs) [12,23–25]. Integration can occur directly into a gene, both introns and exons, and can lead to varying changes in gene expression level[26–28]. In a study of HPV-positive head and neck SCC (HNSCC), 17% of integrants were also found within 20 kbp of a gene, indicative of possible selective pressure from integration at, or near to, coding regions through modifications to host gene expression[29]. Varying explanations for modified gene expression include HPV integration as being commonly associated with amplification of the local region or indeed rearrangement and translocation of that region elsewhere[27,29–34]. Other studies have suggested that higher level transcriptional control may be at play: integration into flanking regions of genes, sometimes as far away as 500kb, has been found correlated with large increases in gene transcription, possibly due to a long-range *cis* interaction between the integrated HPV18 promoter/enhancer and the associated gene[26,27,35,36]. Hence, it is likely that interactions between distant regions of chromatin have ultimately driven selection of this cell line from an excised lesion.

In work presented here, we have used five HPV16 integrant cloned cell lines from the well-established W12 cell model[24,37,38] to investigate, beyond epigenetic/chromatin control of virus early gene expression, further features of genome integration and host gene expression modulation that could lead to the selection of individual cells during carcinogenesis. Using cells that constitute type I integrants (i.e. retain HPV *E6/E7* genes but with deletion or disruption of *E2*) with four or less copies of integrated genomes that have expression per template levels of oncogenes, and which encompass the range seen within the total panel of W12 integrant clones[21], we have developed a novel and state-of-the-art technique (HPV integrated site capture/HISC) to determine HPV16 integration sites genome-wide at nucleotide resolution while utilising HPV16-specifc Region Capture Hi-C to investigate potential interactions between the integrated virus genome and host chromatin. We have been able to pinpoint the precise locations of HPV16 integration sites within our W12 integrant clones, coinciding with areas of open chromatin, as well as determining that integration likely occurs through

microhomology-mediated repair mechanisms. Integration occurs through either a direct process, whereby regions of the host genome are deleted–in our case partially homozygously, with a portion of both alleles within the host break points being deleted–or via a 'looping' mechanism by which flanking regions of the host genome become amplified. Furthermore, application of our Region Capture Hi-C technology has determined that interaction loops do indeed exist between the integrated virus genome and host chromatin both at short- (<500 kbp) and long-range (>500 kbp). Alongside RNA-seq data, we have also confirmed that these interactions do appear to drive host gene dysregulation, possibly through the disruption of usual host: host chromatin interactions within topologically associating domains (TADs), leading to individual gene and cross-region changes in host gene expression during the early stages of cervical neoplastic progression.

## Results

### Integrated HPV16 genomes interact in *cis* with host chromosomes

As it had previously been shown that interactions between an integrated HPV genome and host chromatin on the same chromosome could lead to changes to host gene expression that could be selected for during carcinogenesis[35], we wished to determine whether these *cis* interactions were occurring at an earlier stage of cervical neoplasia in our HPV16-positive W12 integrant clones prior to selection. To address this, we first developed a HPV16-specific Region Capture Hi-C protocol (S1A Fig) similar to that published previously[39,40], here using biotinylated RNA baits specific for the HPV16 genome that would select chimeric DNA complexes based upon the virus sequence from a Hi-C library and hence allow determination of any genome interactions in our five integrant clones previously epigenetically characterized [21]. The regions of significant interaction between the HPV16 and host chromatin were determined using GOTHiC software (with significance determined by cumulative binomial test and the threshold set at <0.05) and visualised using the Circos tool (Fig 1). With all clones, there was an absence of interaction reads between deleted regions of the HPV16 genome but also largely across the *E1* gene due to the necessary length of RNA bait across this region. In W12 clone G2, reads originate from all MboI fragments across the HPV16 genome, indicating interactions with the host and, although the distribution of the reads was fairly uniform, the greatest percentage of reads came from HPV16 gene *E7* (Fig 1A). Significant interactions occurred exclusively with chromosome 5, the chromosome of integration, with the majority likely due to *cis* interactions with bordering host sequences at the breakpoint. However, upon closer inspection of the single chromosome view (Fig 1A inset), there was a divergence between the majority of reads from the virus—indicating the integration site—and a subset of reads that mapped to a separate region of the host, indicating a long-range interaction between the integrated HPV16 genome in G2 and the host. In clone D2 (Fig 1B), the HPV16 genome also interacted with the site of integration at chromosome 5; however, the virus-host reads, predominantly from the *L2* gene, at this scale appear to converge on a single point at the host chromosome (Fig 1B inset). For three further W12 integrant clones (clones H, F and A5), interactions again occurred across the HPV16 genome with the host chromosome of integration, with the majority of reads coming from the *E2* portion of the virus genome for all (Fig 1C–1E). In clone H, the HPV16 genome integrated into chromosome 4 and resulted in a large deletion within the host genome (~170 kbp), which is illustrated by the separation of the virus-host reads in the chromosome view (Fig 1C inset). Interestingly, and contrary to our previous publications[15,21,24], we found that W12 clones F and A5 had the same integration site, with virus-host reads converging on the same region of chromosome 4 (Fig 1D and 1E).

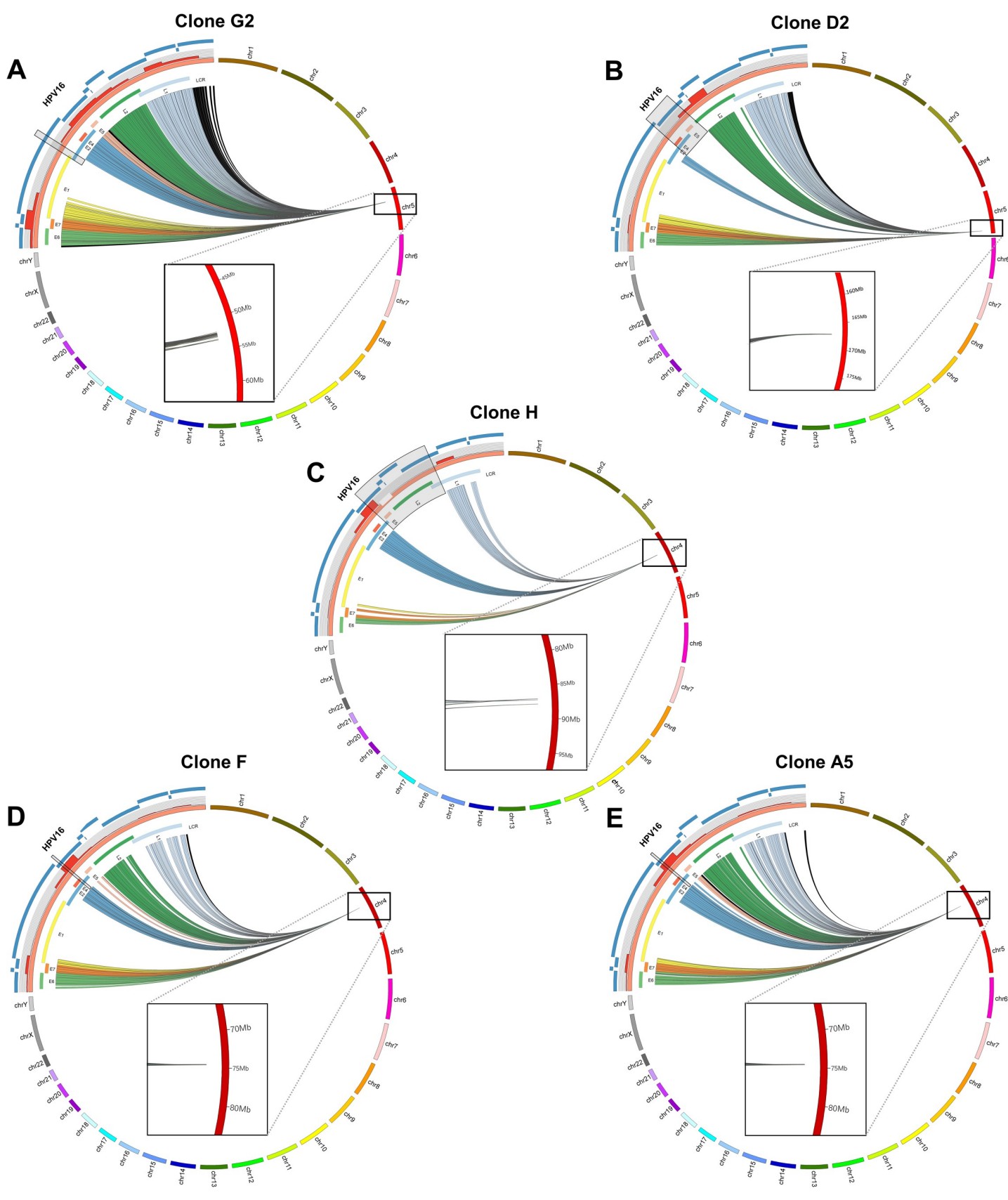

**Fig 1. HPV16-specific Region Capture Hi-C determines definitive HPV16 integration sites.** CIRCOS plots show sequence interactions between HPV16 (orange) and host chromosomes (various) for clones (A) G2, (B) D2, (C) H, (D) F and (E) A5. Each line within the circle represents a significant virus-host read indicating an above background interaction between a region of the HPV16 genome and the host. Reads are coloured to match individual HPV16 genes: E6 = green, E7 = orange, E1 = yellow, E2 = blue, E4 = red, E5 = pink, L1 = dark green, L2 = light blue and non-coding regions = black. Percentage of reads coming from different regions of the virus is indicated by the histogram on the outside of the HPV16 genome, which is split into 500 bp windows (red bars). HPV16 RNA bait fragments used in the Capture Hi-C experiment are indicated on the outside of the CIRCOS plot (blue curved lines); interacting reads are largely absent due to either deletion of the genome region during integration or specifically from the E1 gene due to the necessary length of the bait covering this region (see Materials and Methods section for further information). Grey shaded boxes denote the region of HPV16 genome deleted in individual cell lines. Presented data were generated using the Gothic program and plots are not to scale. Insets show zoomed sites of integration, with interaction divergence in clones G2 and H.

## HPV16 integration site virus-host breakpoint identification at nucleotide resolution

Having developed the probability that virus-host genome interactions did indeed occur soon after HPV16 integration, we next sought to precisely identify the host sequence of the virus-host junctions. To do this with sufficient depth from genomic DNA samples, we developed a novel protocol by which DNA from each integrant clone was enriched for HPV16 sequences along with its flanking regions before sequencing, henceforth known as HPV Integration Site Capture (HISC) (S1B Fig). The resulting sequencing data was analysed for reads mapping to the HPV16 genome with the corresponding human tag being determined. These data were then aligned to the HPV16 genome (S2 Fig) and the human genome (S3–S6 Figs). From all integrant cell lines, sequencing reads mapped to both the HPV16 and host genomes with peaks at only two distinct sites each (S2–S6 Figs, HISC track), regardless of HPV16 genome copy number, demonstrating that only a single 5' and 3' breakpoint existed in each W12 clone examined. The finding of only a single 5' and 3' for each W12 integrant cloned line assessed was confirmed using 'LUMPY', a probabilistic framework for structural variant discovery[41], whereby only four pairs of unique junctions were found with a range of a minimum of 805 split-reads (SRs) (clone H) to a maximum of 6446 SRs (D2). Moreover, the separation of the breakpoint peaks from the HPV16 genome was consistent with termination of RNA-seq reads from separate transcriptome analysis (S2 Fig, RNA-seq tracks) and the known deletion of a proportion of the virus genome in each cell line[21], with the HPV16 transcription profiles additionally being consistent with our previously published quantitative PCR data[15]. Alignment of the breakpoint peaks with host sequence gave a separation distance ranging from ~25–170 kbp across the integrant clones; the separation of peaks here however is consistent with two processes of HPV integration. The greatest distance of 170 kbp seen in clone H occurred due to the deletion of a proportion of the host genome upon 'direct' integration of the single HPV16 genome here determined by quantitative PCR (qPCR) of sections of host genome spanning the integration site (S3B Fig). The breakpoint peak separation distance for the other four clones, however, was due to a 'looping' mechanism of HPV integration[42], whereby a flanking length of host sequence is amplified during integration of one or more copies of the virus. Again, this was verified through qPCR of host DNA across the integration sites (S4B, S5B and S6B Figs), with transcription clearly occurring through these amplified host regions driven from the integrated HPV16 early promoter (S3A, S4A, S5A and S6A Fig). The resultant genomic effect on the structure of both host alleles is pictured in S3C, S4C, S5C and S6D Figs.

To confirm the virus-host breakpoints determined by HISC, PCR across each junction was carried out and Sanger sequenced, with the verified coordinates of breakage summarized in S7 Fig. Interestingly, a 53 bp region of the *E2* gene was found to have been inverted after an intermediate 20 bp deletion, which had not been highlighted by alignment of RNA-seq reads to the HPV16 genome due to bioinformatics processing (S7D Fig). Quantitative PCR (qPCR) was

also carried to show that both the 5' and 3' breakpoints were unique to each cloned line (other than F and A5), with either very low or non-existent products produced with DNA from an early episomal population of W12 cells (W12par1 p12) or a normal cervix line, NCx/6 (S8 Fig).

Identification of the precise virus-host breakpoints in each cloned line allowed the extent of microhomology at integration sites to be assessed. At all integration sites, at least one end of the insertion involved nucleotides from both genomes directly adjacent to the junction being homologous, with a mode of 5 nt (range 3–5 nt) (S9A–S9D Fig), with clone G2 having 5 nt homology at both ends (S9A Fig). Additionally, when microhomology of 10 nt either side of each breakpoint was compared to that generated from 10,000 random shuffles of each sequence extended to 1,000 nt, 4 of 5 integrant clones (G2, H, F and A5) had statistically significant homology at one flank (S9A, S9C and S9D Fig).

## HPV16 integrates into regions of open and transcriptionally active host chromatin

Mapping of the HPV16 integration sites from each cloned cell line allowed investigation of the genomic location and epigenetic landscape into which the virus had inserted (S10 Fig). In 4 of 5 cases, HPV16 had integrated into a gene (D2, *TENM2*; H, *MAPK10*; F & A5; *RASSF6*) with all of these cases arising at introns (S10B–S10D Fig). Despite the integration site in clone G2 occurring intergenically, all insertions occurred at locations of open chromatin, characterized by DNaseI hypersensitivity sites from publicly available normal human epidermal keratinocyte (NHEK) ENCODE datasets (www.encodeproject.org). Additionally, the integration sites showed higher than average levels of histone post-translational modification marks associated with enhancers and transcriptional activity, namely H3K27ac and H3K4me1/2/3, with the marked exclusion of transcriptionally repressed facultative heterochromatin mark, H3K27me3. Noticeably, although integration in clone H did not occur directly into one of these loci, the HPV16 genome is located within 300 kbp of a similar site (S10C Fig). Presence of binding sites of host architectural protein CTCF was found across the sites of integration, although again with higher than average occurrence, and usually within ~20 kbp of the inserted HPV16 genome.

## Short- and long-range *cis* interactions occur between the HPV16 and host genomes

To inspect interactions between the HPV16 and host genomes, Region Capture Hi-C sequencing data was re-visualised using SeqMonk software (Babraham Bioinformatics) and again aligned to NHEK ENCODE datasets at the location of integration for each cloned cell line. Local inspection (~100 kbp window) of interactions between the integrated HPV16 and host genome in clone G2 showed highest levels of interaction corresponding to the 5' and 3' virus-host junctions (Fig 2A, red bars), with above background interactions occurring within this region of host genome amplification (Fig 2A, green-orange bars). This profile was consistently found across the Region Capture Hi-C datasets for all clones (Figs 3A and S11), with the exclusion of clone H where the absence of any intermediate interactions between virus:host breakpoints is likely due to deletion of this host region during integration (S11A Fig).

Upon increasing the window range across the integration loci to ~700 kbp (with re-normalised read depth), peaks of genome interaction outside of the initial 100 kbp window could be seen in clone G2 (Fig 2B) and clone D2 (Fig 3B). Multiple short-range (<500 kbp) interactions were present ranging up to ~240 kbp and ~360 kbp away from the site of integration in clones G2 and D2, respectively (Figs 2B and 3B). As before, LUMPY analysis of HISC sequencing libraries confirmed that true chimeric DNA reads were only found at virus integrant

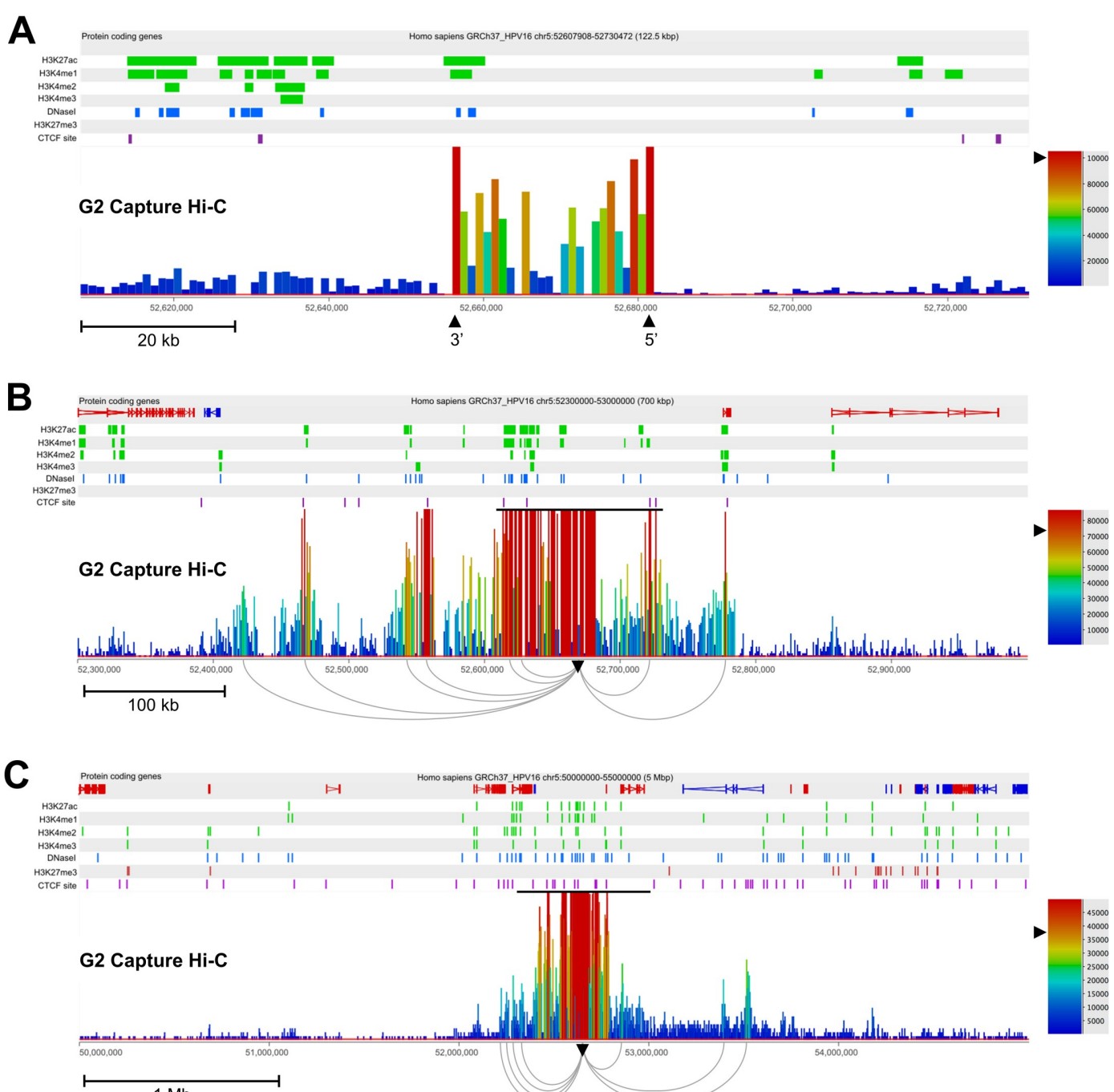

**Fig 2. Identification of short- and long-range interactions between integrated HPV16 genomes and the host chromosome in W12 clone G2.** (A) Capture Hi-C data is presented 122.5 kbp across the HPV16 integration locus. The 5' and 3' breakpoints of the virus genome (which is not aligned here) are indicated by the tallest red bars and are labelled with black arrowheads, being inverted in comparison to the direction of host sequence due to the 'looping' integration mechanism. (B) Capture Hi-C data is presented 700 kbp across the HPV16 integration locus. The black line above the read peaks indicates the genomic window seen in panel A. Peaks of reads indicate regions of the host genome interacting in cis with the integrated virus genome. Short-range interactions between the HPV16 genome and host regions were resolved by consensus using Gothic and are shown beneath the panel, originating from the HPV16 integration site (inverted black arrowhead). (C) Capture Hi-C data is presented 5 Mbp across the HPV16 integration locus. The black line above the read peaks indicates the genomic window seen in panel B. Peaks of reads indicate regions of the host interacting in cis with the integrated virus. Long-range interactions between the HPV16 genome and host regions were resolved by consensus using Gothic and are shown beneath the panel, originating from the HPV16 integration site (inverted black arrowhead). Statistically significant interactions were determined by a cumulative binomial test where adjusted p-value (the q-value), was set a threshold of q<0.05. In each panel, the scale bar represents the normalised read count. Additionally, protein-coding genes are shown in the first track with the direction of each gene indicated by colour (red, forward; blue, reverse), followed by the alignment of ChIP-seq data from the NHEK cell line (ENCODE). Post-translational histone modifications of host

enhancers (H3K27ac, H3K4me1; green), active promoters (H3K4me2, H3K4me3; green), repressed chromatin H3K27me3 (red), DNaseI hypersensitivity sites (blue) and CTCF sites (purple) are shown. Coordinates presented for each window are indicated at the top of each figure.

breakpoints and, therefore, with a very high likelihood (binomial test, p<0.00001) that virus: host looping interactions were not aberrantly called due to covalent linkage between HPV16 and host genomes at further breakpoints. Interestingly, sites of high intensity of interaction with the host genome in both cell lines (Figs 2B and 3B, red bars) overlapped with histone marks of transcriptional activation or enhancers and DNaseI hypersensitivity sites, whereas these sites only correlated with host CTCF binding sites in clone G2 (Fig 2B, purple dashes). The presence of a long-range interaction (>500 kbp) was seen at this scale in clone D2 ~530 kbp from the integration site with the 3' end of host gene *TENM2* (Fig 3B). However, long-range interactions in clone G2 were only visible when expanding the window range to 5Mb,

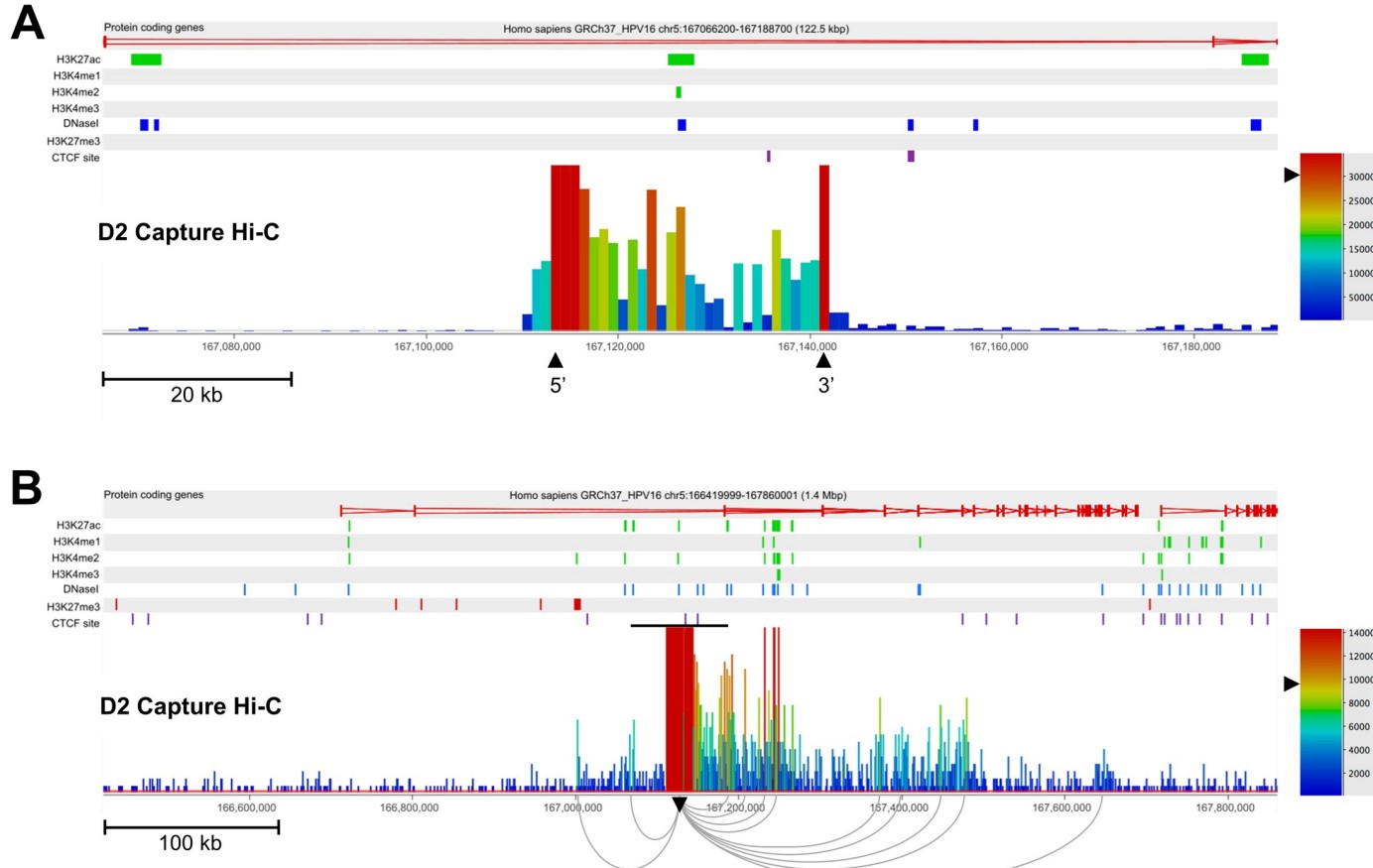

**Fig 3. Identification of short- and long-range interactions between integrated HPV16 genomes and the host chromosome in W12 clone D2.** (A) Capture Hi-C data is presented 122.5 kbp across the HPV16 integration locus. The 5' and 3' breakpoints of the virus genome (which is not aligned here) are indicated by the tallest red bars and are labelled with black arrowheads. (B) Capture Hi-C data is presented 1.4 Mbp across the HPV16 integration locus. The black line above the read peaks indicates the genomic window seen in panel A. Peaks of reads indicate regions of the host interacting in cis with the integrated virus genome. Short-range interactions between the HPV16 genome and host regions were resolved by consensus with Gothic and are shown beneath the panel, originating from the HPV16 integration site (inverted black arrowhead). Statistically significant interactions were determined by a cumulative binomial test where adjusted p-value (the q-value), was set a threshold of q<0.05. In each panel, the scale bar represents the normalised read count. Additionally, protein-coding genes are shown in the first track with the direction of each gene indicated by colour (red, forward; blue, reverse), followed by the alignment of ChIP-seq data from the NHEK cell line (ENCODE). Post-translational histone modifications of host enhancers (H3K27ac, H3K4me1; green), active promoters (H3K4me2, H3K4me3; green), repressed chromatin H3K27me3 (red), DNaseI hypersensitivity sites (blue) and CTCF sites (purple) are shown. Coordinates presented for each window are indicated at the top of each figure.

whereupon two clear interactions were determined downstream of the integration site, with the furthest ~900 kbp from the HPV16 integration site (Fig 2C). This interaction was located at Chr5:53,520,000 within the first intron of host gene *ARL15*, coinciding with a cluster of host CTCF binding sites at a topologically associating domain (TAD) boundary (Fig 2C, purple dashes) and is also the long-range interaction visible on the CIRCOS plot (Fig 1A, inset). Additionally, to statistically confirm that the aggregation of interactions between virus and host genomes could not have occurred by chance, we used a hypergeometric test (Fisher's exact test) to compare the region of interactions in both clones G2 and D2 (Chr5:52,000,000–54,000,000 and Chr5:167,000,000–167,600,000, respectively) to equivalent length, adjacent 'blank' regions (Chr5:50,000,000–52,000,000 and Chr5:166,400,00–167,000,000 for clones G2 and D2, respectively) (Figs 2C and 3B, respectively) that had a level of background reads analogous to those seen across the chromosome, which clearly showed a significant difference (p<0.00001) in both cases.

To validate that direct interaction might occur between the integrated HPV16 genome and the *ARL15* intron in clone G2, three-dimensional (3D) DNA fluorescence in situ hybridisation (FISH) was carried out (Fig 4). Three fluorescent DNA probes were produced to hybridise to either the integrated HPV16 genome (green), the *ARL15* site of interaction (red), or a control region of the host genome (purple) with the same linear distance in the opposite direction (Fig 4A). Only cells containing one HPV16 signal and two copies of both the control and *ARL15* probes were analysed. A representative image from the resulting dataset (30,000 cells) is shown in Fig 4B and analysis of the 3D distances (x, y and z planes) indicated that, on the integrated chromosome, the HPV16 and *ARL15*-specific probes were significantly closer together than the HPV16 and control-specific probes (Fig 4C and 4D). Additionally, it was found that the distance between the control and *ARL15*-specific probes on the integrated allele was significantly lower than that on the unintegrated allele (Fig 4E and 4F). In comparison, the HPV16 signal in W12 clone H was rarely found near to that of either *ARL15* or control region, consistent with the HPV16 integration site being on a different chromosome, whilst HPV16 signal was never found in HPV-negative Raji cells (S12 Fig). Therefore, collectively DNA FISH and capture Hi-C data strongly support a direct interaction between the integrated HPV16 genome and the host *ARL15* gene in clone G2 (Fig 4B, inset).

## HPV16 genome integration sites with short-range (~25kbp) host genome amplification do not affect host topologically associating domain boundaries

Since interactions between integrated HPV16 and host genomes had been confirmed, we next asked whether these interactions could lead to structural changes in local nuclear architecture. Hi-C libraries, containing global host:host interactions, from both clones G2 and D2 were sequenced and, due to the different sites of integration on chromosome 5, these datasets were compared against each other using Insulation Score analysis (Fig 5). This methodology determines how insulated (i.e. isolated from interaction in three-dimensional space) one genomic region is from its neighbouring regions[43] and, as such, results in an albeit lower resolution but more robust analysis of genome interactions across, for example, a 5Mb window. Consequently, regions with the most positive insulation score can be defined as topologically associating domain (TAD) boundaries. Heatmaps of clone G2 Hi-C data (Fig 5A) showed an interaction profile consistent with data from clone D2 (Fig 5B), with interactions occurring across the 5Mb window including clearly defined regions of interacting host DNA up to ~1Mb, approximating to the size of an average mammalian TAD. Insulation score analysis also determined across the 5Mb window no significant change between the two interaction

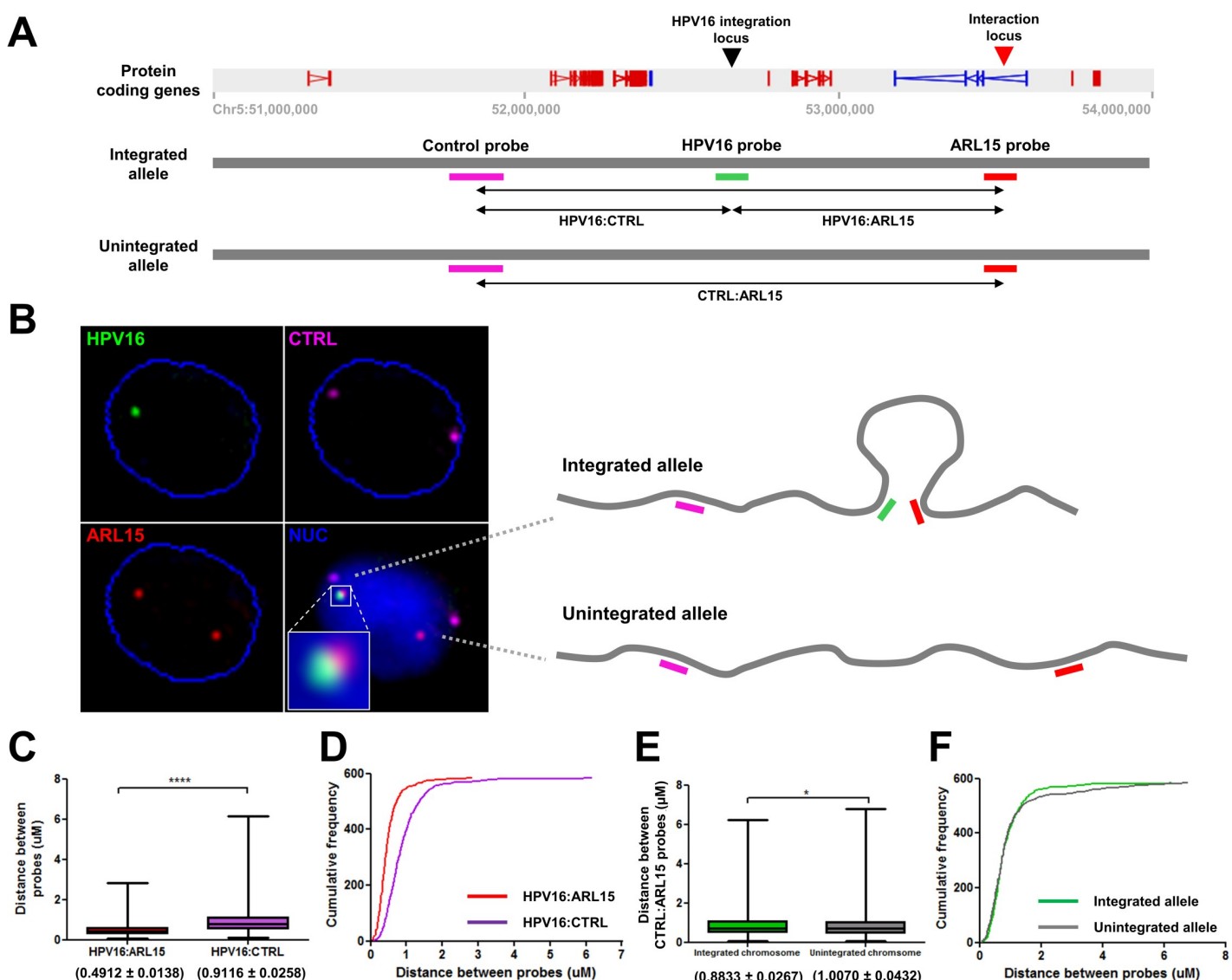

**Fig 4. Validation of HPV16-host genome cis interactions in W12 clone G2 by fluorescence in situ hybridisation (FISH).** (A) Schematic detailing the complementarity of the DNA probes used on the integrated and unintegrated alleles of a portion of chromosome 5 (51–54 Mbp) in W12 clone G2 to confirm interaction between the HPV16 genome (black arrow) and ARL15 gene (red arrow): Control probe (51,676,020–51,873,551; purple), HPV16 probe (green), and ARL15 probe (53,473,886–53,584,235; red). Possible interactions between probe regions are also highlighted. (B) Representative images of the probes hybridised to W12 clone G2 genome of one cell in a 3D FISH experiment (nucleus boundary, blue), a composite image with DAPI (NUC, blue) stain, and interpretation of the associated chromosome spatial conformations. A zoom of the interaction signals is inset. (C) Box-whiskers plot and (D) frequency distribution chart of the distance between both sets of FISH probes in the integrated allele of chromosome 5: HPV16:ARL15 (red box) and HPV16:control (purple box). (E) Box-whiskers plot and (F) frequency distribution chart of the distance between the Control and ARL15 probes in both the integrated (green) and unintegrated (grey) alleles. Lower and upper whiskers denote the 10th and 90th percentiles, respectively, of the distribution. The lower and upper limits of the boxes indicate the 25th and 75th percentiles, respectively. Solid line in the box denotes the median. Numbers below the box plots denote mean ± SEM (n = 585) from which an unpaired, two-tailed Students T-test was conducted; $^{*}p<0.05$, $^{****}p<0.0001$.

profiles despite the presence of HPV16 genomes at the integration site in clone G2 (Fig 5C). Indeed, this finding was replicated when 2.5Mb either side of the clone D2 HPV16 integration site was compared to that of clone G2 (Fig 5D–5F). Hence, HPV16 genome integration and interactions with the host chromosomes did not appear to be affecting the local nuclear architecture at the hierarchical level of TADs and their boundaries.

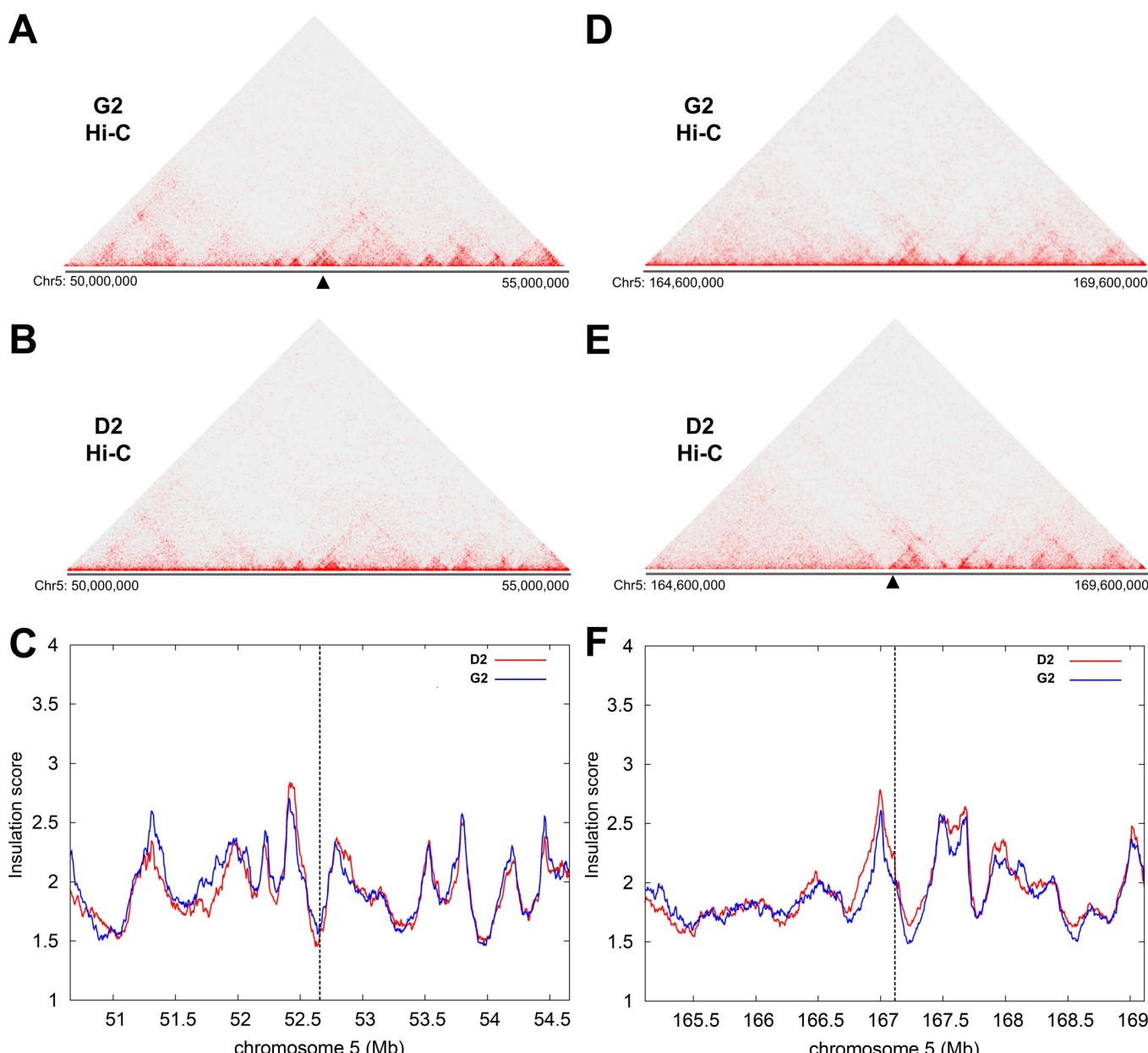

**Fig 5. HPV16 genome integration does not disrupt host topologically associating domain (TAD) boundaries at the integration site in W12 clones G2 and D2.** (A, D) Hi-C data for W12 clone G2 is compared to (B, E) Hi-C data for W12 clone D2 using (C, F) an insulation score interpreting the boundaries of topologically associating domains (TADs) for HPV16 integration sites within W12 clone G2 (Chr5: 50–55 Mbp; left column) and W12 clone D2 (Chr5: 164.6–169.6 Mbp, right column) showing no significant change to either window. Black arrowhead = integration site.

### HPV16 genome integration does modulate intra-TAD host:host genome interactions and host gene expression

As HPV16 genome integration, and resulting interactions with the host genome, did not appear to affect TAD positioning, we sought to determine how far within the host:host chromosome interactions 'loops' from the HPV16 genome occurred. To the existing Region Capture Hi-C datasets, we aligned consensus W12 TAD boundaries (Figs 6 and 7), determined from total Hi-C libraries of clones D2 and G2, which showed no difference in TAD calling

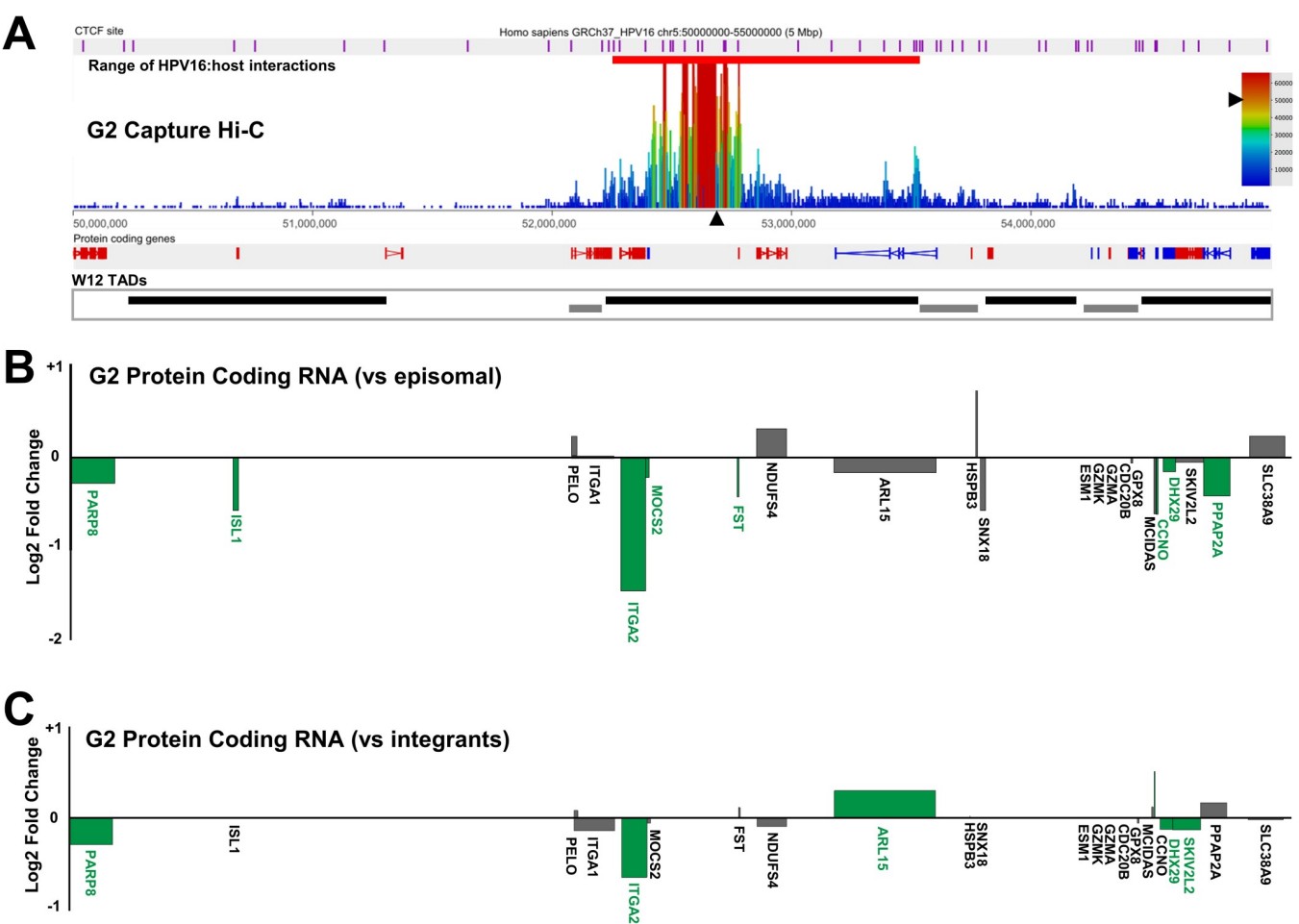

**Fig 6. HPV16 genome integration and virus:host genome interactions lead to significant modulation of host gene expression in W12 clone G2.** (A) Capture Hi-C data is presented across Chr5:50–55 Mbp for clone G2. HPV16 integration site is indicated with a black arrowhead and CTCF sites (purple) aligned across the top of the panel. Aligned protein coding genes are shown in the top track (rightward, red; leftward, blue) with the extent of W12 topologically associating domains (TADs) shown below. Charts indicating the transcript level of host protein coding genes within the 5 Mb region of clone G2 relative to (B) W12 episomal level and (C) mean control level across all other integrant clones. All data is shown as a Log2 fold-change with statistically significant changes indicated by green bars (p<0.05, negative binomial Wald test). Gene length is indicated by width of the corresponding bar.

between clones regardless of HPV16 genome integration, confirming the finding of our insulation score analysis (Fig 5). We found that all HPV16:host genome interactions occurred within, or at the boundary of, the TAD of integration in both clones (Figs 6A and 7A), and all loops in clone D2 occurred exclusively with the *TENM2* gene (Fig 7A).

Next, we addressed whether HPV16 genome integration and interactions with the host chromosome led to any changes in local host gene expression. Transcribed RNAs across a 5Mb window spanning the HPV16 integration site were assessed through comparison of total transcriptome RNA-seq data from an individual integrant clone with transcript levels in both the parental episomal W12 cell line and, separately, an average of all other available datasets from W12 integrant clones with a different integration site. Analysis of protein coding RNAs from clone G2 found that host genes were both up- and down-regulated across the integration locus and in some cases unchanged, with no individual direction of change (i.e. up-regulation or down-regulation) of differential expression restricted to individual TADs (Fig 6B and 6C); a consistent finding across all cloned cell lines analysed. Despite all but one change being less than ±2-fold here (a 2.64-fold decrease in ITGA2 transcript level in comparison to episomal

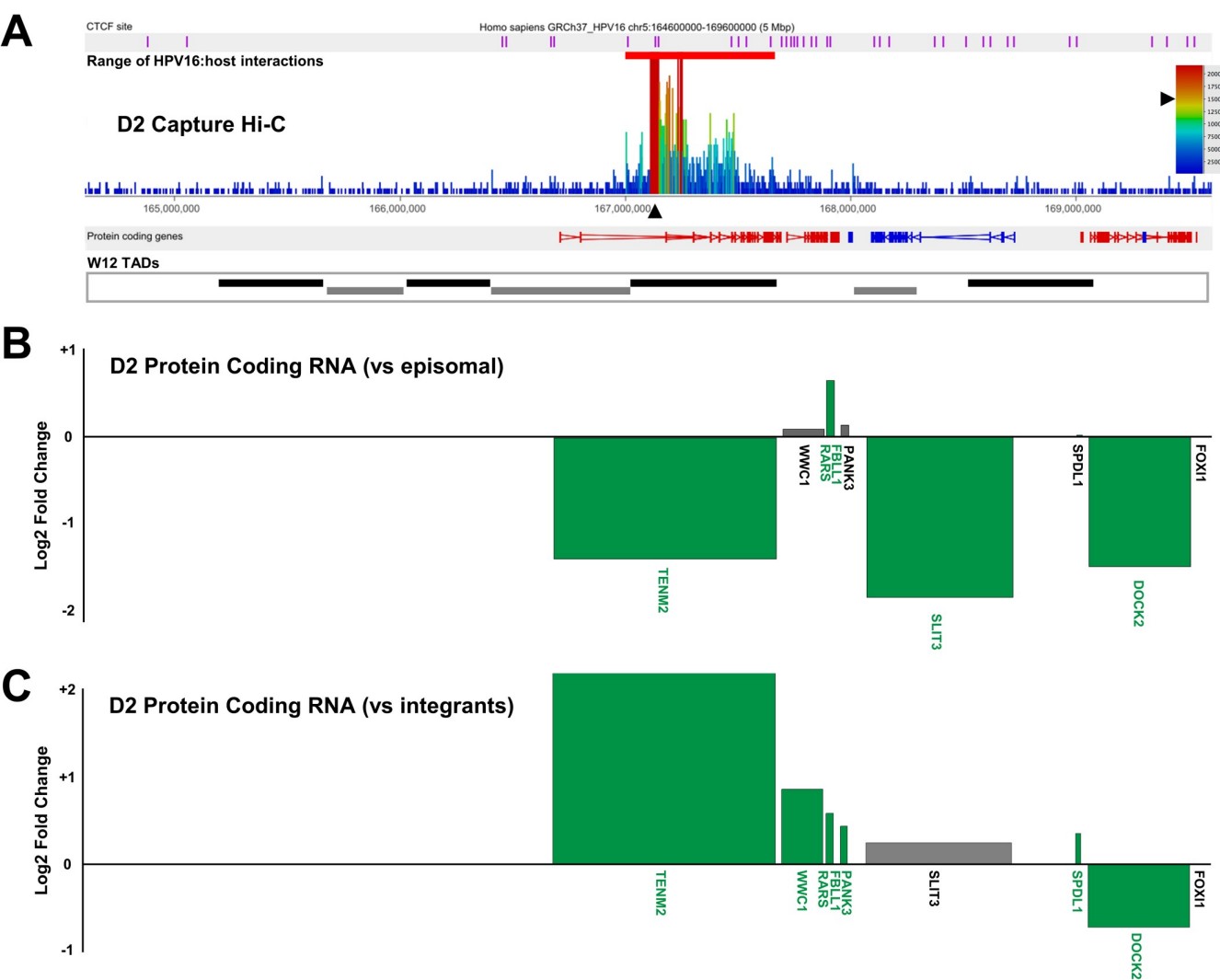

**Fig 7. HPV16 genome integration and virus:host genome interactions lead to significant modulation of host gene expression in W12 clone D2.** (A) Capture Hi-C data is presented across Chr5:164.4–169.6 Mbp for clone D2. HPV16 integration site is indicated with a black arrowhead and CTCF sites (purple) aligned across the top of the panel. Aligned protein coding genes are shown in the top track (rightward, red; leftward, blue) with the extent of W12 topologically associating domains (TADs) shown below. Charts indicating the transcript level of host protein coding genes within the 5 Mb region of clone D2 relative to (B) W12 episomal level and (C) mean control level across all other integrant clones. All data is shown as a Log2 fold-change with statistically significant changes indicated by green bars (p<0.05, negative binomial Wald test). Gene length is indicated by width of the corresponding bar.

cells (Fig 6B)), some genes were found to be significantly down-regulated versus both episomal and mean integrant clone levels (Fig 6B and 6C, respectively; green labelled genes) (*PARP8*, *ITGA2*, *DHX29*, *SKIV2L2*) whilst, interestingly, the only significantly up-regulated gene was that with a confirmed HPV16:host interaction, *ARL15* (1.23-fold; *p*<0.05) in comparison to mean integrant clone levels (Fig 6C). Compellingly, upon comparative analysis of Hi-C libraries between clones G2 and D2 (a more direct comparison of the local interactions than Insulation Score analysis), a decrease in a host:host interaction was found within the TAD of integration in clone G2 (S13A Fig, yellow outlined triangle), in which bounds aligned with the HPV16:host interaction at the TAD boundary within the *ARL15* gene and just downstream of *ITGA2*. Hence, some changes of host gene expression within TADs were found to correlate with modulated host:host interactions as well as novel virus:host interactions.

Analysis of host gene expression at the HPV16 integration site of clone D2 again found that transcription was both up- and down-regulated across the 5Mb region, whilst some genes appeared unaffected (Fig 7). The host gene into which HPV16 had integrated and exclusively interacted with via looping, *TENM2*, was down-regulated in comparison to episomal cells (Fig 7B) likely due to the very high transcript level in parental W12Par1 cells (S14 Fig). However, in comparison to the integrant clone average, TENM2 transcript was up-regulated 4.79-fold (Fig 7C). This highlights the importance of using the mean of several cloned lines as a comparate for modified host transcript levels, in comparison to true low passage W12 episomal cells, as these cloned lines were isolated from the parental population using at least 60 population doublings. No significant changes in host:host interaction were found here within the TAD of HPV16 integration or the adjacent TADs (S15 Fig). Interestingly, all other clones that exhibited HPV16 integration within a host gene showed up-regulation of expression of that gene in comparison both to episomal and integrant clone lines. Expression of *RASSF6* in clones F and A5 increased in both cases ~ 2.3-fold versus episomal cells (Fig 8B and 8D, respectively) and ~1.6-fold in comparison to mean integrant levels (Fig 8C and 8E, respectively), with $p < 0.0001$ in all cases. This was in contrast to the varied up- and down-regulation of expression across the 5Mb window when comparing all other analysed genes in clones F and A5 (Fig 8B–8E, respectively).

In clone H, despite deletion of some of the host coding exons, expression of *MAPK10* was increased 5.06-fold and 4.47-fold ($p < 0.001$ in both cases) versus W12 episomal and mean integrant transcript levels, respectively (Fig 9B and 9C, respectively). Moreover, further interrogation of RNA-seq data determined chimeric HPV16:host RNA-seq reads indicating both breakpoint fusion transcripts and spliced transcripts from the integrated HPV16 genome (including differential HPV16 exon expression) into an adjacent host exon (ENSE00001811960; Chr4:86,952,584), which was the likely cause of the overall increase in expression levels of this gene (S16 Fig). Indeed, splicing events from the integrated HPV16 genome into the host was seen across three of the five integrant clones analysed (S1 Table), including clone G2 where intergenic HPV16 integration led to spliced fusion transcripts with non-coding host DNA (verified through PCR and Sanger sequencing), presumably through cryptic splice acceptor sites (S17 Fig). Interestingly, all splicing events occurred with host DNA within the region of host DNA amplification flanking the HPV16 integration site in clone G2.

## HPV16 genome integration modulates host gene expression across the chromosome

To further investigate the effect of HPV16 integration on host gene expression, the variance in gene expression in the genomic regions adjacent to the HPV16 integration site was compared with that of the whole chromosome. Here, comparing RNA-seq data of 100 genes either side of the integration site (coordinates of genome ranges given in S2 Table) within an individual clone to an average of all other available datasets from W12 integrant clones with a different integration site, host transcript level variance (from both protein coding and non-coding regions) was calculated by grouping five adjacent genes into 'bins' (average genomic range of 1.1Mb ±0.3Mb) and then comparing the range and variance of gene expression within each bin to that of a mean level from across the whole chromosome (Fig 10). Although individual protein-coding genes at HPV16 integration sites had previously shown statistically significant changes in comparison to the integrant clone average (Figs 6C, 7C, 8C, 8E and 9C), averaging transcript levels from both protein-coding genes and non-coding genes led to statistical significance at only the integration site in clone A5 (Fig 10E and 10J). Expression of host genes across all regions indeed appeared variable, however bin mean fold-changes of greater than or equal

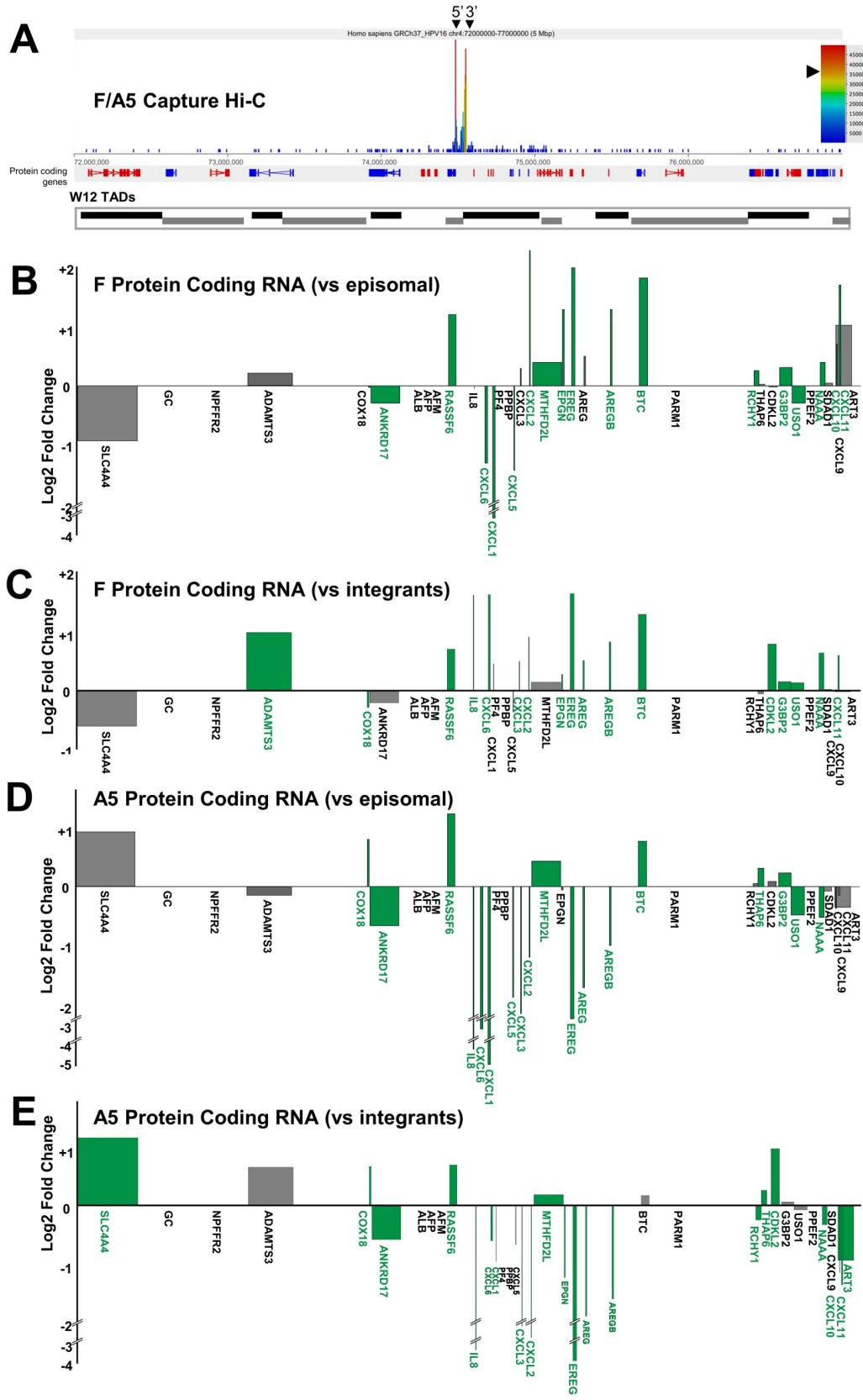

**Fig 8. HPV16 genome integration leads to significant, but differential, modulation of host gene expression in W12 clones F and A5.** (A) Representative Capture Hi-C data is presented across Chr4:72–77 Mbp for clone F and A5. HPV16 integration site is indicated with black arrowheads (breakpoints) and CTCF sites (purple) are aligned across the top of the panel. Aligned protein coding genes are shown in the top track (rightward, red; leftward, blue) with the extent of W12 topologically associating domains (TADs) shown below. Charts indicating the transcript level of host protein coding genes within the 5 Mb region of clone (B-C) F and (D-E) A5 relative to (B, D) W12 episomal level and (C, E) mean control level across all other integrant clones. All data is shown as a Log2 fold change with statistically significant changes indicated by green bars (p<0.05, negative binomial Wald test). Gene length is indicated by width of the corresponding bar.

to 2 in comparison to chromosome mean expression level were limited to only one bin in all clones (Fig 10A–10C and 10E: G2, bin 9; D2, bin -18; H, bin 5; A5, bin 0) except clone F with two adjacent bins (Fig 10D: F, bins -7 and -6). Interestingly, all bins with mean fold-changes of greater than or equal to 2 consisted of up-regulated mean expression (Fig 10A–10D), apart from clone A5 exclusively with down-regulated mean expression at the site of integration (Fig 10E) despite the increase in RASSF6 transcript levels (Fig 8E). Bins at, and adjacent to, the HPV16 integration site were also found to have highly significant gene expression variance (*p*<0.01), although this finding was restricted to integrant clones without virus:host *cis* interactions (Fig 10H–10J). Statistically significant variance was found as far away from the HPV integration site as 5.6Mb in clone H (Fig 10H, bin +5) and 6.4Mb in clone F (Fig 10I, bin -6). Hence, our findings suggest that integration of HPV16 genomes into host chromosomes could have direct modulatory effects on host gene expression beyond both the immediate locus of integration and also the indirect effects of virus oncogene expression on average chromosomal expression.

## Discussion

Progression of disease toward cervical carcinoma is markedly associated with integration of HPV genomes into host chromosomes whereby dysregulation of the control of HPV oncogene expression occurs. Our previous work has shown that, across five integrant cell lines cloned from the W12 model system (F, A5, D2, H and G2), levels of virus transcript per template correlate with multi-layered epigenetic changes that regulate transcription from the integrated HPV16 genomes[15,21]. However, it has not been determined how integration at these sites may produce a more or less selectable clone during outgrowth in our model, which mirrors the natural process of cervical carcinogenesis. This is of particular importance given that integration of HPV16 genomes does not always appear to give these cells a growth advantage beyond the rate of the parental episomal cells[15]. To address this, we developed a state-of-the-art 'HPV integrated site capture' (HISC) technique to determine the precise loci of HPV16 integration sites and also utilised HPV16-specific Region Capture Hi-C to determine if interactions between virus and host genomes could be involved in modulating host gene expression, thereby driving selection of certain clones. With this new technology, we first sought to confirm the integration sites of HPV16 genomes in our five clones through next generation sequencing (NGS) of samples.

In all the W12 integrant clones tested, the HPV16 genome was shown to interact with regions of host chromosomes in *cis*; there were no examples of HPV16 interacting with the host in *trans*, although this could be due to limited sequencing depth. These interactions occurred across the HPV16 genome (unless deleted in an individual clone), with the presence of individual virus genes consistent with the known transcript profiles from the integrated virus genomes in each clone published previously[15]. However, there was an absence of any interactions from a large proportion of the *E1* gene. This was due to technical reasons associated with the design of the RNA baits, based upon *MboI* restriction sites within the HPV16

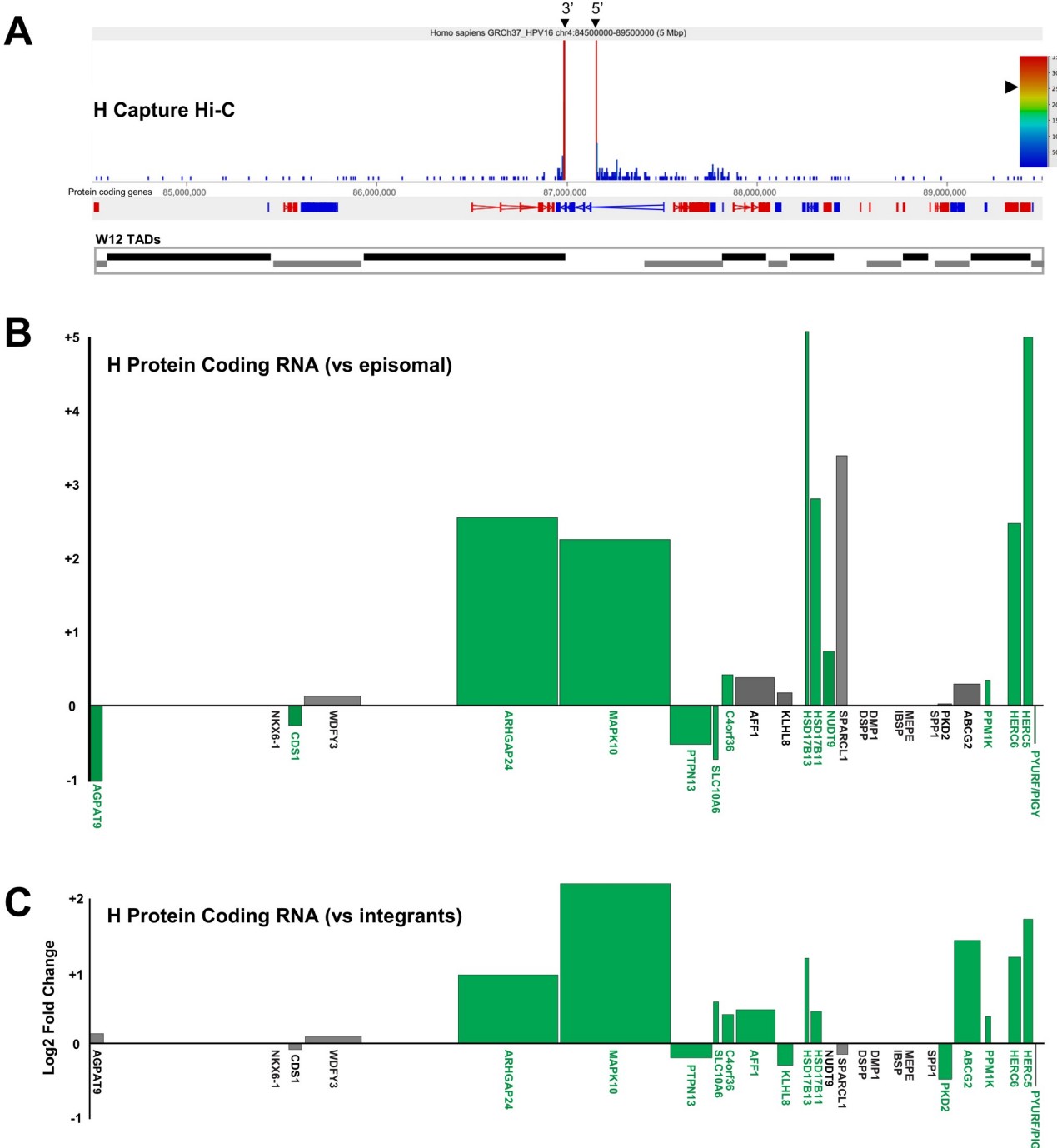

**Fig 9. HPV16 genome integration leads to significant modulation of host gene expression in W12 clone H.** (A) Capture Hi-C data is presented across Chr4:84.5–89.5 Mbp for clone H. HPV16 integration site is indicated with a black arrowheads (breakpoints) and CTCF sites (purple) aligned across the top of the panel. Aligned protein coding genes are shown in the top track (rightward, red; leftward, blue) with the extent of W12 topologically associating domains (TADs) shown below. Charts indicating the transcript level of host protein coding genes within the 5 Mb region of clone H relative to (B) W12 episomal level and (C) mean control level across all other integrant clones. All data is shown as a Log2 fold-change with statistically significant changes indicated by green bars (p<0.05, negative binomial Wald test). Gene length is indicated by width of the corresponding bar.

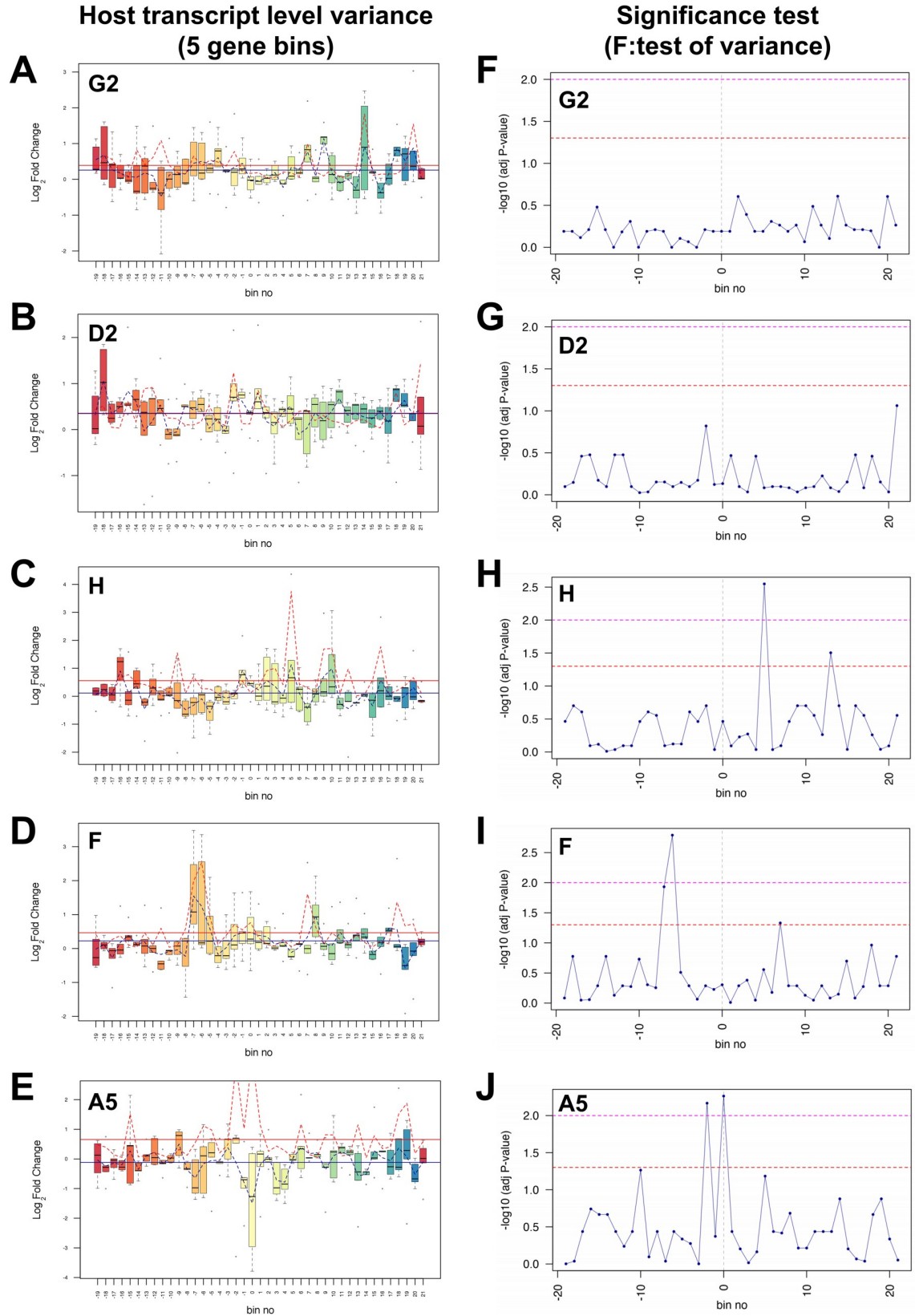

**Fig 10. Variance in host gene expression across the host genomic regions containing the HPV16 integration site in W12 clones.** Each left panel indicates the range and variance of host gene expression in W12 integrant clones (A) G2, (B) D2, (C) H, (D) F and (E) A5, focussing on 100 genes either side of the HPV16 integration site. For each clone, host transcript expression levels were compared with the mean of all other integrant clones. In each panel, the HPV16 integration site is centred on 'bin 0'. Each bin contains five genes, with no overlap between bins. The box and whisker plots illustrate the range of gene expression levels within each bin, with the black bar indicating median values, the box the interquartile range (IQR) and the whiskers the range. The mean gene expression across the whole chromosome is indicated by the solid blue line, while the mean level of gene expression across individual bins is shown by the dotted blue line. The mean variance of gene expression across the whole chromosome is indicated by the solid red line, while the mean variance within individual bins is shown by a dotted red line. All transcript data is Log2 transformed. Each right hand panel shows the significance of the variance (F-test) in gene expression within each bin for clones (F) G2, (G) D2, (H) H, (I) F and (J) A5. Each point represents a five-gene bin, corresponding to those in the left-hand panels. The horizontal lines indicate the significance of the variance in each bin, compared with the variance in gene expression across the whole chromosome (above the dashed red line, $p < 0.05$; above the dashed pink line, $p < 0.01$).

genome, rather than a true biological finding. Additionally, due to the resolution of 1kb used here, our methodology did not support the determination of HPV16:HPV16 loops, which are known to occur in, at least, episomal HPV18 genomes[44].

The determined HPV16:host interactions pinpointed the definitive loci of integration of the HPV16 genomes within each W12 integrant clone, with PCR and sequencing confirming that these locations differed from their original published sites[15,21,24] in all but one clone (clone H). More interestingly, we found that clones F and A5 had the same HPV16 integration site, with the same HPV16 breakpoints and 3' virus genome inversion, likely indicating that one cell line is a precursor of the other, since they exhibit very different phenotypical characteristics[15,21,24]. Our data also support the growing understanding that integration of HPV genomes takes place, at least to some degree, through microhomology-mediated repair (MHMR) of DNA breaks[25,26,29,32,42,45–47]. Our precise determination of HPV16:host breakpoints through HISC, RNA-seq and Sanger sequencing allowed interrogation of the flanking sequences at each junction, showing statistically higher levels of homology between HPV16 and host sequences than expected. Mechanistically, our finding is supported by a recent report that HPV16 E7 protein restrains canonical nonhomologous end-joining (NHEJ) and promotes MHMR, which importantly is associated with a greater sensitivity of HPV-positive HNSCCs to radiotherapy than HPV-negative cancers[48].

Indeed, the evidence for MHMR lends further support to the theory that HPV integration occurs via two main processes: 'direct' and 'looping' integration[32,42]. In our study, mapping of the HPV16 and human DNA recombination junctions support both of these two processes, with subsequent changes to proximal gene expression consistent with effects seen previously [22,32,42]. Furthermore, this precision integration site and breakpoint mapping allowed improved downstream study of virus:host genome interactions. Deletion of a proportion of the host genome adjacent to the HPV16 integration site in W12 clone H is consistent with direct integration and, for the first time as far as these authors are aware, we have shown that deletion can occur in a homozygous fashion to both alleles. All other clones examined (F, A5, D2, G2) showed clear signs of host genome amplification flanking the integration site that would be consistent with looping integration, from which integrants are now sometimes known as 'type III'[22,34]. There remains the possibility that these regions could be amplified as extra-chromosomal virus–host fusion episomes maintained by the HPV origin of replication, which has been proposed following analysis of HNSCC TCGA datasets elsewhere[29,49]. However, our sequencing techniques here appear to consistently illustrate canonical chromosomal integration. Furthermore, the HPV16-specific FISH signal in clone G2 was never seen outside of the signal range for control and *ARL15* gene probes, which might be expected if this amplified region was in some way dislocated from chromosome 5; hence, we conclude that extra-chromosomal units are unlikely.

Despite others' previous analysis of integration sites primarily focussing on cervical SCCs [12], we have also shown in our W12 integrant cell lines, which reconstitute the early stages of cervical carcinogenesis, that HPV16 integration, although occurring in both intronic and intergenic regions in our population, appears more readily to occur (although not with any statistical significance due to the low number of integration sites studied here) within regions of increased chromatin accessibility (DNAseI hypersensitive regions) and with high association of transcriptional enhancers (H3K4me1/H3K27ac) or activity (H3K4me2/3), as found previously[12,50–52]. Intriguingly, an accomplished recent analogous study of one W12-derived subclone (20861), which has 26 tandemly integrated HPV16 genomes interspersed with 25 kb of flanking cellular DNA at chromosome 2p23.2 (a so-called type III integrant), has shown that establishment of super-enhancer like regions can occur through 'looping' integration resulting from amplification of both virus LCR and a basal cellular enhancer. This leads to an enrichment of super-enhancer marks H3K27ac and BRD4, and likely drives high expression of virus *E6/E7* fusion transcripts with subsequent selection and neoplasia[33,34].

Highly specialised Region Capture Hi-C analysis of the W12 integrant clones here confirmed that interactions, both short-range (<500 kbp) and long-range (>500 kbp), between integrated HPV16 genomes and host chromatin, inferred elsewhere previously[22,35], do occur in cell lines that mirror the very early stages of cervical carcinogenesis. We were able to verify that these 'loops' are likely present using DNA FISH analysis in one of our cell lines, clone G2, by confirming the shortened distance between the integrated HPV16 genome and the site of interaction in the *ARL15* gene in comparison to that of a control region. Interestingly, the sites of interaction on the host genome both in clone G2 and D2 more often than expected aligned with consensus sites of interaction of the host architectural protein CTCF; interaction could of course also occur through non-consensus CTCF sites[53,54]. It is known that the HPV genome is able to interact with CTCF through virus-specific sequences[55], contributing to virus transcript levels and splicing[56,57] as well as differentiation-dependent control of virus gene expression through loops with YY1[44]. Thus, it is appealing to hypothesize that insertion of an ectopic CTCF-binding site into the host genome through HPV genome integration, as has been found with the human T-cell lymphotropic (HTLV-1)[58,59] and reported elsewhere for cervical cancers[60], may lead to modulation of the local host genome architecture and changes in host gene expression, which may be naturally selected for. Indeed, even with the loss through integration of the HPV16 *E2* located CTCF-binding site in clone G2, interaction could still be driven by further putative sites in the *L2/L1* genes[55] or could occur through promoter:enhancer-like interactions from the HPV genome, as is believed to be the situation in the HPV18 integrant HeLa cell line[35,36]. This is supported by our finding that loops between the integrated HPV16 genome in our clones also appear to localize to host sites of transcriptional activity and chromatin marks of promoters/enhancers. Whether the orientation of the HPV genome after integration has any effect on directionality of these interactions or loops and indeed whether interaction with these host chromatin domains has any effect on transcript levels produced by the integrated HPV16 genome remains unknown.

Here, HPV16 integration into a gene (*MAPK10* in clone H; *TENM2* in clone D2; *RASSF6* in clones F and A5) clearly caused increases in expression of that host gene in comparison to levels within other integrant clone lines. Despite the loss of some coding exons in *MAPK10* in clone H, direct co-linear insertion of a HPV16 genome led to splicing from the HPV16 genome into the next host exon. Thus, although overall levels of MAPK10 transcript were raised, this was due to an increase in the production of RNA from 3' end exons and possible fusion to HPV16 transcripts. It remains to be determined whether these truncated and/or fusion transcripts could code for protein or indeed whether any expressed protein would be functional.

Regardless of integration mechanism, the presence of at least one integrated copy of an HPV16 genome caused modulation of host gene expression across a wide range of that chromosome. Statistical analysis of groups of host genes, including those at the site of integration, in comparison to other W12 integrant cell lines, showed wide ranging, statistically significant influence on the host gene expression profiles as far away as 6.4Mb (clone F), although fold-changes greater than or equal to two were limited to only one or two gene regions (bins) per clone. Interestingly, a conserved direction of transcript level change within individual TADs (i.e. strictly either up- or down-regulated genes within a single TAD) was not observed here as has been noted in others' previous work[61], although this may be due to the lack of selection used to isolate our early W12 clones here in contrast to studies of clonal carcinomas. Notably, despite the HPV16 integration site in clones A5 and F being identical, due to one likely being a precursor of the other line, other than at the gene of integration (*RASSF6*) local host gene expression profiles contrasted. Interestingly, whilst the transcript level of certain genes shifted in a common direction across both clones, including *ANKRD17* (down), *BTC* (up) and *MTHFD2C* (up), transcript level of the *EREG* gene, well characterised to be associated with numerous cancers[62,63], diverged between clones F and A5. This finding adds further evidence that, as well as differential E6/E7 oncoprotein expression levels and virus genome-associated chromatin modifications due to HPV genome integration[13], the additional effect of host gene expression dysregulation could cumulatively give rise to the disparity in cloned cell growth rates, as seen in our previous publication[15,21].

Our analysis of total host:host genome interactions through interrogation of Hi-C libraries of clones D2 and G2 did not appear to provide evidence that host TADs were greatly affected by HPV16 'looping' genome integration, although it would be useful to corroborate this finding in further clones where integration did not lead to amplification (clones A5 and F) or deletions (clone H). Intriguingly, all of the five integration sites and subsequent modifications to host genome through looping characterised here occur within an individual TAD (clones A5, F, D2 and G2), or within an unclassified region (clone H). All interactions between HPV16 genomes and the host chromosome occurred within the TAD of integration, allowing the hypothesis that TAD boundaries may in some way be able to inhibit HPV:host interactions into adjacent TADs. This barrier to more elongated interactions may also stretch to *trans* interactions with chromosomes other than that of integration, although this remains unverified. Of the clones fully analysed using Insulation Scores (i.e. clones G2 and D2), the range between virus-host breakpoints is relatively close together (~25Kbp) and does not appear to cause vast copy number amplification, as seen in others' work on W12 clones[34] or complex host rearrangements, as has been seen elsewhere in various HPV-associated cancers[32]. Moreover, integration here did not lead to TAD boundary changes, which has been reported in one cervical carcinoma sample leading to enhancer hijacking[61]. It could be postulated that the HPV integration process itself (both 'direct' and 'looping') could be restricted to individual TADs due to the physical nature of the TAD boundaries, and that complex rearrangement beyond these loci and selection might be associated with oncogenic progression. However, this requires much further investigation.

Regardless, interactions between the integrated HPV16 genome in clone G2 and the host *ARL15* gene was associated with a small but statistically significant increase in the level of its transcript. Elsewhere *ARL15* has been found associated as a pro-survival protein, reducing reactive oxygen species (ROS), with both metabolic conditions and kidney cancers[64,65]. Further comparative analysis of our Hi-C data sets showed that this interaction may in fact cause a decrease in the usual host:host intra-TAD interaction between *ARL15* and a region upstream of the HPV16 integration site. Interestingly, the host genes around this upstream interaction site were largely down-regulated, especially the *ITGA2* gene, which has actually

been found over-expressed in certain breast and pancreatic cancers causing increased cancer stemness and metastasis[66,67]. Therefore, it is possible that the relatively stronger interaction between the HPV16 LCR enhancer and *ARL15* gene that might up-regulate ARL15 transcript levels may also supersede usual host interactions, some of which may be promoter:enhancer interactions that maintain regular host transcript levels. Moreover, this modulation of host gene expression distal from the HPV genome integration site is in line with the current theory of Viro-TADs and host gene expression changes[22,59,68]. Indeed, interactions also appear to occur directly between the virus genomes and host enhancers, distinguishable by the host chromatin marks of H3K27ac/H3K4me1 (e.g. clone G2 Chr5:52,545,000 and Chr5:52,775,000), which could inadvertently affect local host gene expression. Thus, whereas host enhancers have been shown to be 'hijacked' through HPV integration, possibly leading to improved competition with other sites[61], our data are consistent with enhancer 'quenching' though destabilisation of regular host:host chromatin interactions, subsequently leading to down-regulation of host transcripts.

Although we are not the first to use Capture technology to precisely determine HPV integration sites[22,26,34,42,46,47,69], our analysis of Region Capture Hi-C and full host:host Hi-C libraries have allowed the first definitive determination of interactions between integrated HPV and host genomes. This had previously been proposed by analysis of 'chromatin interaction analysis with paired-end tag' (ChIA-PET) sequencing data pointing toward a long-range *cis* interaction between the integrated HPV18 promoter/enhancer and the *MYC* gene in the HeLa cell line[35] and may be a possible reason why integration at the 8q24.21 region has been found so often in several studies previously[70–72]. The host genes with which the integrated HPV16 genomes in our cloned lines interact with, namely *ARL15* (clone G2) and *TENM2* (D2), had up-regulated transcript levels in comparison to other integrant clones of a similar passage number, for example TENM2 transcript (4.79-fold), a gene previously associated with cervical neoplasia[73]. However, in the case of ARL15 this change was moderate (1.23-fold). This finding likely highlights that we have isolated HPV16 integrant clones at a very early stage of cervical carcinogenesis with the W12 model system showing the primary effects of HPV16 genome integration; selection of initial integration sites or those that develop either much more elevated expression of oncogenes or supressed levels of tumour suppressor genes due to virus:host interactions could very well occur, which appears to be the case with HeLa cells[35].

In our study, the local (intra-TAD) changes to usual local host:host interactions appear to cause modulation of the host gene expression program, at least within the same TAD. Therefore, integration near to or within a 'cancer-causing gene' does not appear essential to influence such genes due to these *cis*-driven distal events. However, it remains to be fully determined how these HPV:host genome interactions are initiated and maintained, and whether this type of interaction or the downstream effect on host gene expression is selected for, as would be hypothesised through evidence provided by HeLa cells and other cervical carcinomas[22,35,36,61]. Moreover, it remains to be explored whether these virus:host interactions also drive the level of HPV transcripts from the integrated virus genome. Nevertheless, our study provides further insight into early events after HPV16 integration and the mechanisms by which papillomaviruses are able to initiate cervical carcinogenesis.

## Materials and methods

### Cell culture

Detailed descriptions of the W12 system have been published previously[10,74,75], including W12 integrant clone generation[15,24] from the parental, episomal W12 culture series-2 (W12Ser2)[76], whereby HPV16 integrant clones used here more likely represent those that

occur naturally after HPV16 infection due to non-competitive selection, rather than interferon-induced selection pressure used previously[37]. As such, the five W12 clones used here were episome-free, did not express the HPV16 transcriptional regulator E2[15] and were grown in monolayer culture in order to restrict cell differentiation and maintain the phenotype of the basal epithelial cell layer[77]. Additionally, W12 clones were analysed at the lowest possible passage after cloning (typically p3 to p8) in order to minimise any effects of genomic instability caused by deregulated HPV16 oncogene expression[21]. Furthermore, the initial episomal W12 (Par1) cell line[8] and HPV-negative normal cervix line, NCx/6[10,15], were used as additional controls.

### HPV16-host breakpoint and splice junction confirmation

To verify chimeric DNA sequences of HPV16-host breakpoints determined by Capture-seq, and to confirm splice junction sequences from clone G2 RNA-seq analysis, primers were designed using either Primer3 (Primer3Web) alone or Primer-BLAST (NCBI) specific to the DNA sequence (or cDNA sequence from reverse transcribed clone G2 RNA samples (QuantiTect Reverse Transcription Kit, Qiagen)) for PCR (PCR SuperMix High Fidelity, Thermofisher). PCR products were gel extracted and then Sanger sequenced using both 5′- and 3′-end primers to confirm reads from each end of the product. Each analysis was carried out in duplicate. Primer pairs used for PCR of HPV16-host breakpoints and clone G2 splice junctions are given in S3 and S4 Tables, respectively.

### qPCR of HPV16-host breakpoints and genomic DNA

Primers were designed using either Primer3 alone or Primer-BLAST (NCBI) specific to the chimeric DNA sequence of HPV16-host junctions determined by Capture-seq to verify integration sites, as well as host DNA spanning integration sites to determine copy number after HPV16 integration. Primers used for qPCR are given in S5 and S6 Tables. Host DNA copy number was quantified by comparison to TLR2 and IFNβ, as reported previously[15]. Conditions used for all primer pairs on an Eppendorf Mastercycler Realplex were: 95˚C for 2min; 40 cycles of 95˚C for 15sec, 58˚C for 20sec, 72˚C for 15sec, 76˚C for 5sec and read; followed by melting cure analysis from 65˚C to 90˚C to confirm product specific amplification.

### DNA Fluorescence In-Situ Hybridisation (FISH)

BAC clones RP11-467N14 (control locus) and CTD-2015C9 (ARL15 locus) were purchased from (Thermofisher), whereas the HPV plasmid pSP64-HPV16 was prepared in house. BAC/plasmid DNA was purified using the NucleoBond BAC100 kit (Macherey-Nagel), and labelled with aminoallyl-dUTP by nick translation. After purification, 0.5–1 μg labelled BAC DNA was coupled with Alexa Fluor 488, Alexa Fluor 555 or Alexa Fluor 647 reactive dyes (Life Technologies) according to the manufacturer's instructions, and DNA FISH was performed and analysed as described elsewhere[78,79].

### Chromatin crosslinking

Formaldehyde crosslinking of 30 million cells was performed by supplementing standard EGF positive culture medium with formaldehyde to a final concentration of 2% and was carried out for 10 min at room temperature. Crosslinking was quenched by the addition of ice-cold glycine to a final concentration of 125 mM. The adherent cells were scraped from the cell culture plates after crosslinking, collected by centrifugation (400 g for 10 minutes at 4˚C), and washed

once with PBS (50 ml). After centrifugation (400 g for 10 minutes at 4˚C), the supernatant was removed, and the cell pellets were snap-frozen in liquid nitrogen and stored at -80˚C.

## Hi-C library generation

Cells were thawed on ice, and then lysed on ice for 30 minutes in 50 ml freshly prepared ice-cold lysis buffer (10 mM Tris-HCl pH 8, 10 mM NaCl, 0.2% Igepal CA-630, one protease inhibitor cocktail tablet (Roche complete, EDTA-free)). Following the lysis, nuclei were pelleted (650 g for 5 minutes at 4˚C), washed once with 1.25 x NEBuffer 2, and then re-suspended in 1.25 x NEBuffer 2 to make aliquots of 5–6 million cells for digestion. SDS was added (0.3% final concentration) and the nuclei were incubated at 37˚C for one hour (950 rpm). Triton X-100 was added to a final concentration of 1.7% and the nuclei were incubated at 37˚C for one hour (950 rpm). Restriction digest was performed overnight at 37˚C (950 rpm) using 800 units MboI (NEB) per 5 million cells. Restriction fragment ends were filled in using Klenow (NEB) with dCTP, dGTP, dTTP and biotin-14-dATP, and the blunt-ended DNA was ligated following the in-nucleus ligation protocol described previously[80], with minor modifications. Prior to ligation, excess salts and enzymes were removed by centrifugation (600 g for 5 minutes at 4˚C) and the cell pellet was re-suspended in 995 μl of 1 x ligation buffer (NEB) supplemented with BSA (100 μg/mL final concentration). The ligation was carried out using 2000 units of T4 DNA ligase (NEB) per 5 million aliquot of cells, at 16˚C for 4 hours, followed by 30 min at room temperature. Chromatin was then de-crosslinked overnight at 65˚C in the presence of proteinase K (Roche), purified by phenol and phenol-chloroform extractions, precipitated with ethanol and sodium acetate and re-suspended in TLE (10 mM Tris-HCl pH 8.0; 0.1 mM EDTA). The DNA concentration was measured using the Quant-iT PicoGreen assay (Life Technologies). 40 μg of Hi-C library DNA were incubated with T4 DNA polymerase (NEB) for 4 hours at 20˚C to remove of biotin from non-ligated fragment ends, followed by phenol/chloroform purification and DNA precipitation overnight at -20˚C. DNA was sheared to an average size of 400 bp using the Covaris E220 (settings: duty factor: 10%; peak incident power: 140W; cycles per burst: 200; time: 55 seconds). End-repairing of the sheared DNA (using T4 DNA polymerase, T4 DNA polynucleotide kinase, Klenow (all NEB)) was followed by dATP addition (Klenow exo-, NEB) and a double-sided size selection using AMPure XP beads (Beckman Coulter) to isolate DNA ranging from 250 to 550 bp. Biotin-marked ligation junctions were immobilised using MyOne Streptavidin C1 Dynabeads (Invitrogen) in binding buffer (5 mM Tris-HCl pH 8.0, 0.5 mM EDTA, 1M NaCl) and after stringent washing in the same buffer at 55˚C for 10 min ligated to the custom capture-seq adapter using 1600 units of T4 DNA ligase (NEB) for 2 hours at room temperature. The immobilised Hi-C libraries were amplified using the custom primers PE PCR 1.0.33 and PE PCR 2.0.33 with 7–9 cycles. After PCR amplification, the Hi-C libraries were purified with AMPure XP beads (Beckman Coulter). Quantity and integrity of the Hi-C libraries was determined by Bioanalyzer profiles (Agilent Technologies).

## Genomic DNA library generation

Cells were thawed, lysed and nuclei were isolated as described above. Nuclei from 5–6 million cells were treated with SDS and Triton X-100 as described for the generation of Hi-C libraries. All Hi-C specific steps, such as MboI digestion, restriction fragment end fill-in, blunt end ligation and the removal of biotin from un-ligated restriction fragment ends were mock performed by replacing the respective enzymes with an equal amount of water. All other steps were performed as described for the generation of the Hi-C libraries. The biotin-streptavidin pull down was omitted and, therefore, the ligation of the custom sequence adapters was done

in solution by adding 4 µl adaptors (30 µM) and 1600 units T4 DNA ligase (NEB). The ligation was carried at for 2 hours at room temperature on a rotating wheel in 1x ligation buffer (NEB). Pre-capture PCR amplification was carried out using the custom primers PE PCR 1.0.33 and PE PCR 2.0.33 with 7–8 cycles. The amplified libraries were purified with AMPure XP beads (Beckman Coulter) and their quantity and the quality were assessed by Bioanalyzer profiles (Agilent Technologies).

### Capture RNA bait library design

120-mer capture RNA baits were bioinformatically designed to both ends of MboI restriction fragments overlapping the HPV16 genome. Requirements for target sequences were as follows: GC content between 25% and 65%, no more than two consecutive Ns within the target sequences, and maximum distance to a MboI restriction site 330 bp. For short MboI fragments, where 120-mer RNA baits originating from both ends would have overlapped (potentially interfering with optimal hybridization to Hi-C libraries), only the Watson (coding or sense) strand was used for capture RNA bait design, and if necessary the baits were trimmed to minimum length no shorter than 97 nt. This resulted in the design of 16 RNA bait sequences (S7 Table) covering the MboI restriction fragment ends of the entire HPV16 genome, with the exception of two fragments too short (18 and 63 bp, respectively) for capture RNA bait design.

### Biotinylated RNA bait library for Region Capture Hi-C generation

The process of HPV16-specific Region Capture Hi-C was carried out essentially as previously published[81]. DNA sequences encoding for the 16 RNA bait sequences, with different restriction enzymes sites on each side (BglII on one site and either HindII or SpeI on the other), which were separated by a 3 bp random spacing sequence, were ordered as two gBlocks (Integrated DNA Technologies) and cloned into plasmid vectors using the Zero Blunt TOPO cloning kit with One Shot TOP10 Chemically competent cells according to manufacturer's instructions. Both gBlocks were extracted from plasmid DNA by EcoRI (30 units) restriction enzyme digestion at 37˚C for 2 hours. Having a BglII and another restriction enzyme sites on the other end, enabled BglII side specific ligation of a T7 promoter sequence adapter with a BamHI overhang essentially as per manufacturer's instructions, preventing the generation of overlapping complementary transcripts. These adapters were generated by annealing T7_promoter_adapter_1 and T7_promoter_adapter_2 as per manufacturer's insctructions. Digestion with both restriction enzymes and adaptor ligation were done in one reaction simultaneously, in the presence of BamHI. Two reactions containing 700 ng of gBlock1 DNA or 850 ng of gBlock2 DNA, 30 units BglIl each, 100 units BamHI each, 5-fold molar access of pre-annealed T7 promoter adapters and either 80 units HindIII (NEB) or 40 units SpeI (NEB) were incubated at 37˚C for 2 hours in 1x T4 DNA ligase buffer (NEB). Following this incubation 1200 units T4 DNA ligase (NEB) were added to each reaction and incubated at 25˚C for 3 hours. The samples were then run on a 1% agarose gel and specific bands at 180 bp were cut out and gel purified. Equimolar amounts of sequences were *in vitro* transcribed according to manufacturer's instructions using the T7 MegaScript kit (Ambion) with biotin-labelled UTP (Roche). The RNA was then purified using the MEGAclear kit (Ambion) following the manufacturer's instructions.

### Biotinylated RNA bait library generation for HISC

Four consecutive and non-overlapping fragments from the pSP64 HPV16 bacterial artificial chromosome were PCR amplified using the Expand High Fidelity PCR system (Roche). This resulted in complete coverage of the HPV16 genome. T7 promoter sequences (Roche) were

added to one side of the PCR product during this PCR amplification, enabling subsequent directional *in vitro* transcription. Sequences were *in vitro* transcribed in the presence of biotin-UTP and purified as described above, followed by fragmentation to 120 nt with 4 nM magnesium chloride at 95˚C for 7 min. Fragmented biotinylated RNA was purified by Isopropanol precipitation followed by two subsequent washes with 75% ethanol.

## Solution hybridization Region Capture of Hi-C libraries

500 ng to 2000 ng of Hi-C library DNA or genomic DNA library were concentrated using a vacuum concentrator (Savant SPD 2010, Thermo Scientific) and then re-suspended in 5 μl dH$_2$O. 2.5 μg mouse cot-1 DNA (Invitrogen) and 2.5 μg sheared salmon sperm DNA (Ambion) were added as blocking agents. To prevent concatemer formation during hybridization, 1.5 μl blocking mix (300 μM) was added (equimolar mix of P5_b1_for_33, P5_b1_rev_33, P7_b2_for and P7B2_rev (see list X for sequences)). Biotinylated RNA baits were used in a ratio 1:12 to Hi-C libraries (25 ng biotinylated RNA baits per 300 ng of Hi-C library) and supplemented with 30 units SUPERase-In (Ambion). Biotinylated RNA baits for capture DNA-Seq were used in a ratio of 1:3.33 (300 ng RNA baits per 1,000 ng genomic DNA library) and supplemented with 30 units SUPERase-In. The DNA was denatured at 95˚C for 5 min in a PCR machine (PTC-200, MJ Research; PCR strip tubes (Agilent 410022)) and then incubated with the biotin capture RNA at 65˚C, in hybridization buffer (5 x SSPE (Gibco), 5 x Denhardt's solution (Invitrogen), 5 mM EDTA (Gibco), 0.1% SDS (Promega)) for 24 hours, in a total reaction volume of 30 μl. Captured DNA/RNA hybrids were enriched using Dynabeads MyOne Streptavidin T1 beads (Life Technologies) in binding buffer (1 M NaCl, 10 mM Tris-HCl pH 7.5, 1 mM EDTA) for 30 minutes at room temperature. After washing (once in wash buffer 1 (1 x SSC, 0.1% SDS) for 15 minutes at room temperature, followed by three washes in wash buffer 2 (0.1 x SSC, 0.1% SDS) for 10 minutes each at 65˚C), the streptavidin beads (with bound captured DNA/RNA) were re-suspended in 30 μl 1 x NEBuffer 2. Post-capture PCR amplification was carried out using between six to nine cycles using primer pairs that consisted of one TruSeq adapter reverse compliment and the TruSeq universal adapter from streptavidin beads in multiple parallel reactions, which were then pooled to purify the PCR products using AMPure XP beads (Beckman Coulter).

## Paired-end next generation sequencing

Two biological replicates for Hi-C and capture Hi-C libraries were prepared for each of the cell lines, although sequencing of Hi-C libraries was only conducted on W12 clones with statistically significant virus:host interactions beyond the virus:host breakpoints (i.e. clones D2 and G2). Sequencing was performed on Illumina HiSeq 2500 generating 50 bp paired-end reads (Sequencing Facility, Babraham Institute). CASAVA software (v1.8.2, Illumina) was used to make base calls and reads failing Illumina filters were removed before further analysis. Output FASTQ sequences were mapped to the human reference genome (GRCh37/hg19) containing the HPV16 genome as an extra Chromosome and were filtered to remove experimental artefacts using the Hi-C User Pipeline[82].

## HiCUP and SeqMonk

Sequence data was obtained from Illumina HiSeq paired-end sequencing. Using the HiCUP Pipeline[82] paired-end Capture Hi-C (cHi-C) fastq files were mapped with Bowtie 2[83] to a human GRCh37 reference containing a HPV16 pseudo-chromosome. HiCUP removes invalid and artefactual di-tags by overlaying the di-tags on an *in silico* restriction digest of the reference. The resulting BAM files contained putative di-tags for use in subsequent analyses.

SeqMonk (https://www.bioinformatics.babraham.ac.uk/projects/seqmonk/) was used to quantitate and visualise the density of di-tags contained in the BAM files. The HPV16 sequence and annotation files were downloaded from the European Nucleotide Archive (www.ebi.ac.uk/ena/data). ENCODE Annotation for NHEK [84] was obtained from Ensembl release 75 [85].

## Circos

The raw cHi-C fastq files were converted to fasta format and BLAST [86] was used to search the HPV16 genome for reads mapping to it. The partner human reads were determined and the 2 sets of reads were mapped to the GRCh37 reference containing the HPV16 pseudo-chromosome using Bowtie 2. The BAM outputs were converted to BED format and modified to be compatible with the circular visualisation tool Circos [87]. The HPV16 genome was split into bins of 500 bp and the count per bin determined from the chimeric human-HPV16 di-tags. The counts, the HPV16 MboI restriction map and gene coordinates were annotated on the Circos plots.

## GOTHiC

The HiCUP output was converted to format compatible with the Bioconductor package GOTHiC[88]. To find significant interactions between distal locations GOTHiC implements a cumulative binomial test based on read depth. This was used to identify regions of the human genome in contact with the HPV16 pseudo-chromosome at a resolution of 1kb. Statistically significant interactions were determined by a cumulative binomial test; results were filtered on their adjusted p-value (the q-value), where the threshold in this case was set at <0.05. As such, q-values ranged anywhere between 0.049 to 3.043e-13. Di-tag mappings were visualised with Circos after filtering the previous Circos input by the GOTHiC determined interactions.

**Breakpoint mapping with USearch.**   The precise sites of HPV16 integration in the W12 cell lines were identified by sequencing HISC libraries. The raw fastq files were converted to fasta format and BLAST was used to search for reads mapping to the HPV16 genome. From these, the corresponding human tags were determined. Fast clustering of the reads with USearch[89], based on an sequence identity score of 0.65, identified clusters of sequences in the human and HPV16 derived reads. Consensus sequences from non-singleton clusters were obtained by aligning the clustered reads to each other using Clustal Omega[90]. The breakpoints were inferred from these consensus sequences and validated by Sanger Sequencing[91]. From the validated integration sites, custom chimeric references were generated for each W12 line. Due to the existence of tandem amplifications in some of the regions of integration, two versions of the chimeric human-HPV16 chromosomes were generated. In the first case, the HPV16 provirus was 5' of a single amplified human sequence. For the second, the provirus was placed 3' of the amplified human sequence. For another W12 line, 'H', there is a deletion in the region of integration and this was reflected in the chimeric chromosome.

**Breakpoint confirmation with LUMPY.**   DNA-seq read libraries were aligned to a custom GRCh37 reference containing an additional HPV16 pseudochromosome with BWA--MEM (v0.7.17) (https://arxiv.org/abs/1303.3997). Alignments were sorted with SAMtools (v1.10)[92] and PCR duplicates and other artefactual reads identified using Picard's MarkDuplicates tool (v2.20.1) (http://broadinstitute.github.io/picard). HPV16 integration breakpoints were identified from split and discordant alignments using the probabilistic, structural variant (SV) discovery tool, LUMPY (v0.3.1)[41] with its default parameters. Breakpoint calls that failed LUMPY's default filtering and those with less than 500 split-reads (SRs) per junction were filtered out using BCFTools (v1.10.2)[93].

**Juicer and juicebox.** Using the specific chimeric references, Hi-C contact maps at different resolutions were generated from the available raw Hi-C fastq files for clones G2 and D2 using the Juicer Pipeline[94]. Juicer constructs a compressed contact matrix from pairs of genomic positions located in close proximity in 3D space. The Hi-C contact maps were imported into Juicebox[95] for visualisation.

**HOMER.** The 'findTADsAndLoops' function from the HOMER suite[96] of NGS analysis tools (v4.11) was used to detect topologically associating domains (TADs) at 5kb resolution in 25kb windows and calculate insulation scores. 2D TAD annotation was imported into Juicebox with no difference being seen between Hi-C libraries for clones G2 and D2 at the respective sites of HPV16 integration.

## Insulation score plots

Measurement of the topological domain structure along the chromosomes was computed with an average insulation score profile at the TAD boundaries. The insulation score is the standardized -log enrichment of contacts between the downstream and upstream 300kb regions ($-\log(a/(a+b1+b2))$) where 'a' is the number of contacts between, and 'b1' and 'b2' the number of contacts within the upstream and downstream 300kb regions. Using this definition, a more positive insulation score indicates a stronger TAD boundary.

## RNA-seq analysis and alignment

Complementary DNA (cDNA) libraries were prepared for the five W12 clones (F, A5, D2, H, G2) epigenetically characterised previously[21], two further clones with low HPV16 copy number, distinct integration sites and statistically differing growth rates versus parental lines (R2, 10q22.1, insignificant; B, 8q24.21, significant)[15] to give greater power to statistical analysis here, and the original episomal W12 (Par1) cell line (two biological replicates each) by total RNA extraction from confluent cells with Ribo-Zero rRNA depletion and DNAse treatment before cDNA was prepared with the TruSeq RNA and DNA Sample PrepKit (Illumina). 50bp paired-end cDNA libraries sequenced on an Illumina HiSeq 2000 (Genomics Core Facility, EMBL Heidelberg). Sequence adapters were trimmed from the reads with Kraken[97]. Trimmed FASTQs were mapped against a GRCh37.p13 reference transcriptome (Ensembl version75) that included HPV16 transcript annotation using STAR[98] with default parameters. Strand specific gene counts were obtained from alignments with HTSeq[99] and differential gene expression analysis performed using the R/Bioconductor package DESeq2[100]. Modulation of host transcript levels due to virus integration was then evaluated per clone in comparison to the mean expression of all other clones with RNA-seq libraries (i.e. versus a 6-clone mean expression level), and additionally in comparison to the original episomal W12 cell line. Variance significance (false discovery rate (FDR) of 0.05) was calculated using an 'F-test of equality of variances' with a Benjamini Hochberg adjustment of P-values by FDR multiple testing correction.

## Supporting information

**S1 Fig. Diagram illustrating the principles of 'HPV16-specific Region Capture Hi-C' and 'HPV Integration Site Capture' (HISC).** (A) HPV16-specific baits (consolidated pictorially as black line) are used to isolate both short- and long-range interactions (red double headed arrow) between a capture region (integrated HPV16 genome) and the host genome. (B) HPV16-specific baits (black lines) are used to enrich HPV16:host breakpoints from an 'undigested' sequencing library. (blue indicates host chromosome, genes and promoters.) (TIF)

**S2 Fig. W12 integrant clone HPV16 genome length and breakpoint determination by 'HPV Integration Site Capture' (HISC) and RNA-seq analysis.** Aligned to a cartoon of the HPV16 genome are DNA read peaks from HISC analysis (grey peaks) determining break-points and deleted region of the HPV16 genome (red underline) as well as RNA read peaks (blue) denoting transcription across the virus genome for each of the W12 integrant clones used in the study along with RNA reads for the parental, episomal W12 Par1 (Epi) cell line for comparison.
(TIF)

**S3 Fig. Arrangement of genomic DNA at clone H integration site.** (A) RNA-seq data (blue peaks) showing transcription from host sequences driven by the integrant HPV16 genome (red arrow indicates the direction of which HPV16-host read-through transcription occurs through breakpoint) and HPV Integration Site Capture (HISC) data (multi-coloured peaks due to base calls) verifying virus-host breakpoints on the host genome. Wild-type allele regions indicated below with approximate location of qPCR primer sites. (B) qPCR of genomic DNA regions to determine copy number after HPV16 genome integration in clone H. (C) Determination of arrangement of gDNA sections after HPV16 genome integration through 'direct' mechanism, causing deletion of regions including homozygous deletion of region C. Virus copy number (V, 1) from Scarpini et al., 2014. Not to scale.
(TIF)

**S4 Fig. Arrangement of genomic DNA at clone G2 integration site.** (A) RNA-seq data (blue peaks) showing transcription from host sequences driven by the integrant HPV16 genome (red arrow indicates the direction of which HPV16-host read-through transcription occurs through breakpoint) and HPV Integration Site Capture (HISC) data (multi-coloured peaks due to base calls) verifying virus-host breakpoints on the host genome. Wild-type allele regions indicated below with approximate location of qPCR primer sites. (B) qPCR of genomic DNA regions to determine copy number after HPV16 genome integration in clone G2. (C) Determination of arrangement of gDNA sections after HPV16 genome integration through 'looping' mechanism, amplifying region B. Virus copy number (V, 3) based on Scarpini et al., 2014. Not to scale.
(TIF)

**S5 Fig. Arrangement of genomic DNA at clone D2 integration site.** (A) RNA-seq data (blue peaks) showing transcription from host sequences driven by the integrant HPV16 genome (red arrow indicates the direction of which HPV16-host read-through transcription occurs through breakpoint) and HPV Integration Site Capture (HISC) data (multi-coloured peaks due to base calls) verifying virus-host breakpoints on the host genome. Wild-type allele regions indicated below with approximate location of qPCR primer sites. (B) qPCR of genomic DNA regions to determine copy number after HPV16 genome integration in clone D2. (C) Determination of arrangement of gDNA sections after HPV16 genome integration through 'looping' mechanism, amplifying region B. Virus copy number (V, 4) from Scarpini et al., 2014. Not to scale.
(TIF)

**S6 Fig. Arrangement of genomic DNA at clones F and A5 integration sites.** (A) RNA-seq data (blue peaks) showing transcription from host sequences driven by the integrant HPV16 genome (red arrow indicates the direction of which HPV16-host read-through transcription occurs through breakpoint) and HPV Integration Site Capture (HISC) data (multi-coloured peaks due to base calls) verifying virus-host breakpoints on the host genome. Wild-type allele

regions indicated below with approximate location of qPCR primer sites. qPCR of genomic DNA regions to determine copy number after HPV16 genome integration in clones (B) F and (C) A5. (D) Determination of arrangement of gDNA sections after HPV16 genome integration through 'looping' mechanism, amplifying region B. Virus copy number (V, 1) from Scarpini et al., 2014. Not to scale.
(TIF)

**S7 Fig. Summary of virus-host junction genomes at HPV16 integration sites.** In all schematics, host chromosomal DNA is shown in blue and the orientation indicated by the grey arrow above (5' to 3'). Integrated HPV16 DNA is shown in orange, with the viral oncogenes and long control region (LCR) highlighted in red, and the direction of transcription from the viral early promoter shown by an arrow from the LCR. The location of the viral breakpoint in base pairs is given above the junction, whereas the cellular DNA breakpoint in base pairs is given below the junction. The genome copy number and length of the integrated HPV16 genome is indicated in red above the schematic. When HPV16 has integrated into a host gene, the orientation of this gene is shown beneath the schematic in black. (A) W12 clone G2, (B) D2, (C) H and (D) W12 clones F and A5. (Virus genome copy number taken from Scarpini et al., 2014.)
(TIF)

**S8 Fig. Confirmation of HPV16-host breakpoints by quantitative PCR.** To confirm the virus-host breakpoints found in each W12 integrant cloned line, pairs of qPCR primers were designed to amplify the 5' (left column) and 3' (right column) breakpoints from genomic DNA samples for clones (A) G2, (B) D2, (C) H, (D) F and (E) A5 in comparison to the episomal (W12par1) cell line and HPV-negative cell line (NCx/6).
(TIF)

**S9 Fig. Regions of HPV16 and host sequence homology at the integration site.** Figures show comparisons between the virus-host sequences obtained by Sanger sequencing and the normal host and HPV16 genomic sequences, 25 nucleotides either side of the breakpoint (indicated by a central dotted line). HPV16 DNA sequence = black, inverted HPV16 DNA sequence = green, human DNA sequence = blue, homologous nucleotides = red. Significant levels of microhomology between host and HPV16 sequences were calculated by comparing the homology seen at the 10 nt directly either side of the breakpoint compared to 1000 nt of extended sequence which was shuffled 10,000 times and are indicated by a line above appropriate sequences. *p<0.05 (highlighted in red).
(TIF)

**S10 Fig. Analysis of host chromatin structure at the sites of HPV16 integration.** Each panel shows 5 Mb of the host genome across the integration loci for W12 clones (A) G2, (B) D2, (C) H, and (D) F/A5 with the virus integration site indicated by a black arrow. Protein coding genes are shown in the first track and the direction of each gene indicated by colour (red, forward; blue, reverse). ChIP-seq data from a normal human epidermal keratinocyte (NHEK, taken from ENCODE) cell line is aligned with the host genome. Post-translational histone modifications of host enhancers (H3K27ac, H3K4me1; green), active promoters (H3K4me2, H3K4me3; green), repressed chromatin (H3K27me3, red), DNaseI hypersensitivity sites (blue) and CTCF sites (purple) are shown. Coordinates presented for each clone are indicated at the top of each figure. W12 clone H has both 5' and 3' ends of the HPV16 genome identified due to the length of the host genomic deletion.
(TIF)

**S11 Fig. HPV16-host breakpoints identified in clones H, F and A5 by HPV16-specific Region Capture Hi-C.** (A) Capture Hi-C data is presented 515 kbp across the HPV16 integration locus for W12 clone H. The 5' and 3' breakpoints of the virus are indicated by the tallest red bars and are labelled with black arrowheads, running leftward due to the direction of virus sequence and without intermediate reads due to deletion of host sequence during 'direct' integration mechanism. (B) Capture Hi-C data is presented 200 kbp across the HPV16 integration locus for W12 clone F and (C) for W12 clone A5. The 5' and 3' breakpoints of the virus are indicated by the tallest red bars and are labelled with black arrowheads, being inverted in comparison to the direction of host sequence due to the 'looping' integration mechanism. In each panel, the scale bar represents the normalised read count. Additionally, protein-coding genes are shown in the first track, followed by the alignment of ChIP-seq data from the NHEK cell line (ENCODE). Post-translational histone modifications of host enhancers (H3K27ac, H3K4me1; green), active promoters (H3K4me2, H3K4me3; green), repressed chromatin (H3K27me3, red), DNaseI hypersensitivity sites (blue) and CTCF sites (purple) are shown. Coordinates presented for each window are indicated at the top of each figure.
(TIF)

**S12 Fig. Fluorescence in situ hybridisation (FISH) control data for validation of three-dimensional chromatin interactions in W12 clone G2.** Representative images of probes (used for assessment of W12 clone G2 cells in Fig 4) hybridised to the genome of one (A) W12 clone H cell (n = 1327) and one (B) Raji cell (n = 1284) in a 3D FISH experiment. Probes target: HPV16 (green); Control region (CTRL) (Chr5: 51,676,020–51,873,551; purple); and ARL15 (Chr5: 53,473,886–53,584,235; red). Nuclear boundary is defined as blue circle and a composite image (bottom right) is present with DAPI (NUC, blue) stain. The single HPV16 integration site in W12 clone H is present on chromosome 4 (Chr4: 86,983,196–87,153,458, with host deletion), whereas Raji cells are HPV-negative.
(TIF)

**S13 Fig. Chromosomal integration of HPV16 genomes causes host transcript modulation through host:host interaction changes within TADs.** (A) Comparative analysis of Hi-C libraries between clones G2 and D2 shows significant differentially interacting regions (DIRs) within the TAD of integration (red–increased host:host interactions in clone G2 vs D2; blue–decreased host:host interactions in clone G2 vs D2), including the decreased interaction in clone G2 between the *ARL15* locus/TAD boundary (Chr5: 53.52Mbp, turquoise arrowhead) and the locus between *MOCS2* and *FST* (Chr5: 52.4Mbp, orange arrowhead), highlighted as a 'yellow outlined triangle'. HPV16 integration site is indicated with a black arrowhead. Data is aligned to sites of host CTCF interaction (purple lines). (B) Associated Capture Hi-C data is presented across Chr5: 51.5–54 Mbp. HPV16 integration site is indicated with a black arrowhead (scale bar represents the normalised read count). (C) Aligned protein coding genes (rightward, red; leftward, blue) and the extent of W12 topologically associating domains (TADs) are shown below. Charts are presented indicating the transcript level of host protein coding genes within the 2.5 Mb region of W12 clone G2 relative to (D) W12 episomal (Par1) levels and (E) the mean transcript levels of all other integrants used in the study. All data is shown as a $Log_2$ fold change with significant changes ($p<0.05$) indicated by green bars. Gene length is indicated by width of the corresponding bar.
(TIF)

**S14 Fig. Host transcript level at gene of HPV16 integration sites across W12 integrant clones and episonal cells.** Heatmap of normalised transcript counts from RNA-seq data of the host genes at which HPV16 integration occurred in clones D2 (*TENM2*), H (*MAPK10*), F/A5

(*RASSF6*) and the gene of virus:host interaction in clone G2 (*ARL15*).
(TIF)

**S15 Fig. Chromosomal integration of HPV16 genomes causes host transcript modulation without host:host interaction changes within TADs.** (A) Comparative analysis of Hi-C libraries between clones D2 and G2 shows no significant host:host interaction changes within the TAD of integration (red–increased host:host interactions in clone D2 vs G2; blue–decreased host:host interactions in clone D2 vs G2); aligned to sites of host CTCF interaction (purple lines). HPV16 integration site is indicated with a black arrow. (B) Associated Capture Hi-C data is presented across Chr5: 166.5–168 Mbp. HPV16 integration site is indicated with a black arrow (scale bar represents the normalised read count). (C) Aligned protein coding genes (rightward, red; leftward, blue) and the extent of W12 topologically associating domains (TADs) are shown below. Presented charts indicate the transcript level of host protein coding genes within the 1.5 Mb region of W12 clone D2 relative to (D) W12 episomal (Par1) levels and (E) the mean transcript levels of all other integrants used in the study. All data is shown as a Log2 fold change with significant changes (p<0.05) indicated by green bars. Gene length is indicated by width of the corresponding bar.
(TIF)

**S16 Fig. Integration of HPV16 genomes into host chromosomes in W12 clone H causes virus:host spliced fusion transcripts.** (A) Diagram summarises HPV16:host breakpoints and deletion of introns/exons in *MAPK10* gene with (B) spliced fusion transcripts found by RNA-sequencing in W12 clone H.
(TIF)

**S17 Fig. Integration of HPV16 genomes into host chromosomes in W12 clone G2 causes virus:host spliced fusion transcripts.** (A) Diagram summarises HPV16:host fusion transcripts found by RNA-sequencing in W12 clone G2. (B) Mapping of the fusion transcript HPV16 splice donor locations and host splice acceptor sites. (C) Determination of G2-specific fusion transcript expression levels via qPCR in comparison to total E6 coding transcripts (E6all) in clones G2 (blue), A5 (orange) and episomal W12par1 (grey).
(TIF)

**S1 Table. RNA-seq fusion reads including HPV16 sequence from W12 cells.**
(TIF)

**S2 Table. Coordinates of the genomic range of 100 genes flanking HPV16 integration sites analysed for transcript level variance in Fig 10.**
(TIF)

**S3 Table. Primers for PCR amplification and Sanger sequencing of HPV16-host break-points.**
(TIF)

**S4 Table. Primer pairs for PCR and qPCR amplification of clone G2 HPV16:host spliced RNAs.**
(TIF)

**S5 Table. Primers for qPCR amplification of HPV16-host breakpoints.**
(TIF)

**S6 Table. Primers for qPCR amplification of host gDNA.**
(TIF)

**S7 Table. MboI restriction sites and RNA baits for HPV16-specific genome cleavage and capture.**
(TIF)

## Acknowledgments

We thank Dr Olga Mielczarek for assistance with FISH analysis.

## Author Contributions

**Conceptualization:** Ian J. Groves, Nicholas Coleman.

**Formal analysis:** Ian J. Groves, Emma L. A. Drane, Marco Michalski, Jack M. Monahan, Cinzia G. Scarpini, Stephen P. Smith, Giovanni Bussotti, Csilla Várnai, Stefan Schoenfelder, Anton J. Enright.

**Funding acquisition:** Ian J. Groves, Nicholas Coleman.

**Investigation:** Ian J. Groves, Emma L. A. Drane, Marco Michalski.

**Methodology:** Ian J. Groves, Emma L. A. Drane, Marco Michalski, Jack M. Monahan, Cinzia G. Scarpini, Stephen P. Smith, Giovanni Bussotti, Csilla Várnai, Stefan Schoenfelder.

**Supervision:** Ian J. Groves, Stefan Schoenfelder, Peter Fraser, Anton J. Enright, Nicholas Coleman.

**Visualization:** Ian J. Groves, Emma L. A. Drane, Marco Michalski, Jack M. Monahan.

**Writing – original draft:** Ian J. Groves, Emma L. A. Drane.

**Writing – review & editing:** Ian J. Groves, Marco Michalski, Jack M. Monahan, Stephen P. Smith, Giovanni Bussotti, Stefan Schoenfelder, Nicholas Coleman.

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
