## [Decision Letter · Decision Letter 0]

22 Mar 2021

Dear Dr Groves,

Thank you very much for submitting your manuscript "Three-dimensional interactions between integrated HPV genomes and cellular chromatin dysregulate host gene expression in early cervical carcinogenesis" for consideration at PLOS Pathogens. As with all papers reviewed by the journal, your manuscript was reviewed by members of the editorial board and by several independent reviewers. In light of the reviews (below this email), we would like to invite the resubmission of a significantly-revised version that takes into account the reviewers' comments.

All three reviewers appreciated strengths of the manuscript and found the study to be of interest and significance, but also pointed out important areas to be addressed in revision. Please pay close attention to the points by each of the reviewers, which will be important to address in your point-for-point rebuttal. The reviewers each point out a number of areas that will be important to address, including the importance of revising the writing in a number of places to improve clarity, soften/qualify conclusions that are speculative, to better showcase new vs old findings, and to better discuss effects of how host genome structural alterations at HPV integration sites might also contribute to local nuclear architecture and gene expression profiles. As pointed out by reviewer #1, important areas for a revision should include re-analysis of the significance of the observed looping between viral and cellular sequences, to address whether 2-fold changes in host gene expression result in biologically significant phenotypes and to thoroughly discuss this point; re-analyzing in Fig. 2b which genes are significantly changed in expression in the areas with high levels of looping. As commented on by reviewer #2, an important control is missing in Fig. 4, and higher resolution images should be provided and 4B should be changed to improve clarity. The Dixon et al. TAD calls should be reprocessed to match their own W12 subclone Hi-C data, and RNAseq analysis should be done to compare profiles of expression changes within TADS/chromosomes relative to that of W12 parental subclone. As pointed out by Reviewer 3, data should be provided to show growth advantage of integrant clones relative to cells with episomal HPV DNA, to provide more evidence that associated human DNA is statistically significantly above background or covalently joined to HPV DNA with mappable junctions. Likewise, evidence of statistical significance should be added to Figs. 2,3, 6-8 and technical issues with the Fig. 4 FISH analysis should be addressed.

We cannot make any decision about publication until we have seen the revised manuscript and your response to the reviewers' comments. Your revised manuscript is also likely to be sent to reviewers for further evaluation.

Sincerely,

Benjamin E Gewurz, M.D., Ph.D.

Guest Editor

PLOS Pathogens

Karl Münger

Section Editor

PLOS Pathogens

Kasturi Haldar

Editor-in-Chief

PLOS Pathogens

orcid.org/0000-0001-5065-158X

Michael Malim

Editor-in-Chief

PLOS Pathogens

orcid.org/0000-0002-7699-2064

All three reviewers appreciated strengths of the manuscript and found the study to be of interest and significance, but also pointed out important areas to be addressed in revision. Please pay close attention to the points by each of the reviewers, which will be important to address in your point-for-point rebuttal. The reviewers each point out a number of areas that will be important to address, including the importance of revising the writing in a number of places to improve clarity, soften/qualify conclusions that are speculative, to better showcase new vs old findings, and to better discuss effects of how host genome structural alterations at HPV integration sites might also contribute to local nuclear architecture and gene expression profiles. As pointed out by reviewer #1, important areas for a revision should include re-analysis of the significance of the observed looping between viral and cellular sequences, to address whether 2-fold changes in host gene expression result in biologically significant phenotypes and to thoroughly discuss this point; re-analyzing in Fig. 2b which genes are significantly changed in expression in the areas with high levels of looping. As commented on by reviewer #2, an important control is missing in Fig. 4, and higher resolution images should be provided and 4B should be changed to improve clarity. The Dixon et al. TAD calls should be reprocessed to match their own W12 subclone Hi-C data, and RNAseq analysis should be done to compare profiles of expression changes within TADS/chromosomes relative to that of W12 parental subclone. As pointed out by Reviewer 3, data should be provided to show growth advantage of integrant clones relative to cells with episomal HPV DNA, to provide more evidence that associated human DNA is statistically significantly above background or covalently joined to HPV DNA with mappable junctions. Likewise, evidence of statistical significance should be added to Figs. 2,3, 6-8 and technical issues with the Fig. 4 FISH analysis should be addressed.

Reviewer's Responses to Questions

**Part I - Summary**

Reviewer #1: Groves et al. describe a Hi-C analysis of HPV 16 positive cell lines that contain integrated copies of HPV 16 derived from a parental W12 line derived from a biopsy of a precancerous lesion. The authors conclude that HPV 16 genomes integrated into host chromosomes in regions of open chromatin and short duplications or deletions occur at regions flanking the integrated genomes. Hi-C analysis indicates looping between viral sequences and cellular sequences that appear to primarily localized with in the same TAD that contains the integrated viral genome. Interestingly, integration does not lead to changes in expression of other genes in the TAD greater than 2-fold exclusive of the gene proximal to the integration site . While the studies are of interest, a number sections of the manuscript are not clearly written and need to be improved. The significance of the looping studies needs to be discussed

Reviewer #2: The authors present a novel ‘HPV16-specific Region Capture Hi-C’ technique to dissect viral-host chromatin interactions at HPV integration sites and correlate this with local changes in gene expression profiles. They show that both short and long-range three-dimensional (3D) interactions between the integrated viral genome and host chromosomes can modulate host gene expression changes through the disruption of local host:host 3D interactions known as topologically associating domains (TADs) and suggest this as a potential mechanism for clonal selection of HPV integrants at the early stages of cervical neoplastic progression. Using state-of-the-art techniques, these findings extend on earlier work in HeLa cells that have eluded to long-range interactions at the HPV integration site at the MYC locus and provide valuable insight into the role and mechanism of HPV integration in neoplastic progression. The data presented gives a very detailed analysis of viral-host interactions profiled in several subclones of the HPV16-positive W12 cell line, however, more discussion of the effects of associated host genome amplifications, deletions or rearrangements at the integration locus in the context of local nuclear architecture and gene expression profiles would really benefit this manuscript.

Reviewer #3: This paper presents the use of a well-established cervical cancer model (W12) for studying HPV, integration and biological outcomes. The authors isolated clones of integrated HPV, and identified HPV DNA junctions with human DNA. Five clones with HPV-human DNA junctions were isolated, two of which were derived from one parental cell with a recombination event between HPV and human DNA. The authors propose that there is physical association between integrated HPV DNA and human DNA at distal locations on the same chromosome via a looping process, and that genes in the vicinity of the integrated HPV DNA are preferentially deregulated.

There is some confusion with the term “3-D interactions”. This term is used extensively and seems to refer to interactions between DNA of different genomic locations, or between chromatin. Yet, it’s safe to assume that reviewers will regard all interactions within the cell as being 3-D. It’s hard to even think of an interaction that would not be in 3-D. If this term means something more than the obvious then it was not explained sufficiently in the paper.

It is being proposed that HPV DNA integration and resultant altered gene expression results in a growth advantage over cells with episomal HPV DNA. However, data is not presented as to what growth advantage these integrant clones have over non-integrated cells. These clones could easily not have any growth advantage and thus it could suggest that HPV DNA integration or altered gene expression does not offer a consistent mechanism for growth advantage. This is an important point that seems to be skipped over. It should be noted that there can be substantial clonal variation from a single clone for growth rate and any other homeostatic process, such as gene regulation. Therefore, differences in measurements from the 5 clones isolated may only reflect clonal variation unrelated to HPV integration. What is missing in general is analysis of cells with episomal HPV DNA. It’s difficult to study the effects of HPV integration without studying control cells that don’t have HPV integrated.

One approach used for studying integrated HPV DNA interacting with nearby human DNA uses chromatin digested into short DNA sequences and crosslinked such that human DNA located far from the HPV DNA will not be isolated with HPV DNA unless it is crosslinked due to non-covalent interaction. The concept that integrated HPV DNA is uniquely associated non-covalently with distal human DNA (hundreds of Kb away) seems to come from a study of HeLa cells. However, HeLa has the HPV18 genome so scrambled that HPV DNA can be found recombined with human DNA throughout the region, and it’s far from clear from that study that distal human DNA is connected to HPV DNA by non-covalent means in addition to covalent means. The possibility the authors in this paper are addressing of specific interaction between HPV DNA/chromatin and nearby human DNA/chromatin is intriguing. However, the evidence the authors’ present to support this possibility is not sufficiently convincing. First, readers would need to see evidence that the associated human DNA is not merely non-specific background DNA that comes through in the purification. Non-random patterns can appear in the non-specific DNA due to PCR/HT sequencing artifacts and due to DNA abundance variation, which is known to happen near HPV integration sites. It would be good to see a representative region of the human genome that’s not near the integration site to assess what a lack of associated DNA looks like. Quantitation of the associated DNA in a region near the integration site being statistically different from all other regions would be required to show that it’s significant. It’s also necessary to rule out the likelihood that the proposed HPV associated human DNA is covalently joined to HPV DNA with mappable juctions. Just sequencing purified HPV DNA and aligning all reads to the human genome to see if this DNA is the same human DNA found in the crosslinked chromatin study would help address this likely scenario.

It would be helpful in each section that it be made clear when new general information is being presented and what information confirms what has been found in other published studies. For example, mapping of HPV and human DNA recombination junctions and the affected HPV and human genes has been done before quite a bit, so what does the data in this mapping study tell us that’s different. It might not be different and the mapping is just the needed preparation for the subsequent studies and that could be clearly stated.

Data is shown how the HPV integration sites and associated human DNA correspond to chromatin markers in the five clones. Figures 2 and 3 do not present any obvious association between the HPV-associated DNA or integration site and the various chromatin markers based on visual interpretation. Others have found a statistically significant association between integration sites and open chromatin and or active genes, as the authors are suggesting in this data, but those other studies analyzed a large number of integration sites to have sufficient power to establish statistical significance. No statistical analysis was done here and the four integration sites in these five clones may not be of sufficient number to reach statistical significance. There’s also a bit of a scientific leap to think that the chromatin marker patterns in the epithelial cells from the ENCODE analysis would look like the pattern within the W12 cells. A study of the patterns in unintegrated W12 versus once HPV had integrated at these sites might have provided new information.

In Figure 2 and 3, it’s not clear where within the genome map is the integrated HPV DNA in one of more copies. If the HPV genomes occupied space in the map then it would not be possible to align the reference genome with markers and genes. If the HPV genomes were designated by the insertion symbol of a triangle, it was not apparent. All that’s shown is the sites for the two ends of the HPV DNA but no HPV DNA.

FISH signals for probes of HPV and distal human DNA were detected in interphase nuclei as a way to assess physical interaction between the DNA locations. There are problems with this analysis. For interphase nuclei, a general rule for being able to resolve two discreet signals (a distance of around 200 nm or higher) of probes less than 40 Kb indicates a genomic linear distance of at least 50 kb between the two probes. The data in Figure 4 indicate inter-probe distances are non-zero and measurable. So any association does not represent physically contact or the values would be zero. Maybe the median included measurements of un-resolved distances between probe signals. Distances in interphase nuclei between probes varies for a variety of reasons and typically is a crude and not definitive measure of how close two probes are in the genome. The presented interphase nuclei probings merely shows that the HPV probe is closer to the ARL15 probe that it is to the control probe. For the integrated site, it looks like the ARL15-HPV probe distance is about half the ARL-control probe distance, which is what the presented genomic map shows. This would represent no physical association between ARL15 and HPV. Other issues are the use of BACs as probes which cover a 100Kb plus distance that will appear not as a spherical signal but an elongated signal or even a squiggle, which makes assessing separation between probes more difficult. Also, use of non-flattened nuclei and 3-D imaging, such as what confocal microscopy offers would allow for more accurate measurements instead of a 3-D nuclei imaged in 2-D.

Unfortunately, the FISH image was obscured by the colored plus signs and the actual signals are not visible, so it wasn’t possible to assess the image. It wasn’t clear why the summary of data was shown by box plots and not plotting means and standard errors. It wasn’t clear how the p-values were calculated; non-parametric for box plots or parametric for means?

There’s no evidence presented that demonstrates that the HPV DNA and human DNA joined to it is within the human genome as proposed. How do the authors know that the HPV plus ajoined human DNA is not excised and extrachromosomal and replicating autonomously, such as the case with the c-myc gene in cancers unrelated to HPV. A FISH probing of metaphase chromosomes would address this.

For the data in Figures 6 and 7, it would be nice to see that there are flanking regions on both sides of the integration sites in which there’s no gene expression modulation or show a separate representative region with no gene expression modulation. There also needs to be statistical analysis showing if the correlation between integration site and local genes modulated in expression is significant.

For the clones in Figure 7, A5 and F, these are the two clones derived from the same parent cell with the HPV integration event. The gene expression pattern of genes local to the integration site appear to be unrelated. This makes the gene expression analysis for all clones highly suspect. Is the difference in gene expression between the two subclones related to clonal variation?

Figure 8 shows the variance in gene expression as a result of a specific HPV integration event over a 30-50 mb distance covering 200 genes around the integration site with some statistically significant variance yet not for the genes at the site of integration. Why aren’t the genes near the integrated HPV showing statistically significant variance as expected from the gene expression studies in Figure 6 and 7? The clone values are compared to a 6-clone average; what 6 clones would that be? Could the significant variance in gene bins be due to a single gene in each bin? It does not appear that there was any correction for multiplicity in the statistical analysis for the 200 genes. After this reviewer’s correction, none of the significant p-values would appear to be statistically significant.

What does the term “Mio” used multiple times mean? Million? It should be fixed.

**Part II – Major Issues: Key Experiments Required for Acceptance**

Reviewer #1: 1). What is the significance of looping between viral and cellular sequences if only moderate changes in expression of 2-fold cellular genes are observed? A more thorough discussion of this point would be helpful.

2). Different W12 derived clones exhibit looping between regions of the viral genome and cellular sequences. For Hela cells this looping has been shown to lead to enhanced MYC expression. For the high level looping shown in Figure 2B and C (greater than 25K reads) are there genes like MYC located in those loci that are also significantly enhanced or is expression is it at the 2-fold threshold? In either case it would be useful to describe the genes in these loci whose expression is altered.

3). Many of the reads in Hi-C data that are below 500 are unlikely to be significant and only those that are significantly higher are more likely to be physiologically significant.

4). Sequencing of HPV integration sites has been described in many previous studies. Have microhomology regions at junctions of viral and cellular sequences been previously described?

5). Line 303: the authors state there are not consensus CTCF sites present in these loci, but looping could also be due to non-consensus sites

6). Line 339: what is meant by no change in nuclear architecture? What parameters are used and how is this quantified? What is an insulation score?

7), pg. 5 the statement that the integrated clones generated in this tissue culture model are representative of what occurs in vivo progression to cancer is highly speculative and should be identified as such. Have these integrated clones been derived following interferon selection?

8). Line 405: too strong a conclusion

9). Is there any looping within a single integrated viral genome such as the CTCF site in E2 with the YY1 site as described previously?

9). A number of sections are poorly written with lots of jargon making it hard to understand and should be made clearer.

a). 289-294

b). 335-337

c). 351-352

d). 365-369

Reviewer #2: Point 1: For Figure 4, as a control, the authors should include FISH analysis in a HPV-negative cell line to control for HPV16 DNA probe background signal and also repeat in a different W12 subclone to show that the interaction between the integrated HPV16 DNA and ARL15 gene is specific to clone G2. A higher resolution image of the FISH signal should be used in Figure 4B. The coloured “+” symbols indicating the probes for the integrated verses non-integrated alleles are covering the FISH signal and should either be removed and replaced with arrows marking the integrated/non-integrated alleles or made smaller and positioned adjacent to the FISH signal

Point 2: It is stated in the Methods section that Hi-C was performed for all the W12 subclones. For Figure 6, in addition to TADs profiled in Dixon et al., TADs should be called in all W12 subclones for alignment with integration breakpoints in the matched samples to determine whether any of the integration sites disrupt TAD boundaries and consequently gene expression profiles across neighbouring TADs. The authors should reprocess the Dixon et al. TAD calls to match their own Hi-C data in the W12 subclones so that the same parameters for calling TADs across these different cell lines are used for comparison.

Point 3: For analysis of gene expression changes within TADs and across each chromosome, comparison to the RNA-seq profile in the W12 parental (Par1) subclone that maintains extrachromosomal genomes should be included

Reviewer #3: (No Response)

**Part III – Minor Issues: Editorial and Data Presentation Modifications**

Reviewer #1: (No Response)

Reviewer #2: The introduction is very detailed and should be condensed. For example, how the W12 subclones that were used in this study were generated should be briefly described in the methods section and the discussion of the MYC locus should be moved into the final paragraph of the discussion section

Line 153 of the introduction; authors should define type I integrants

Line 171 of the introduction; describe what is meant by ‘partially homozygously’ deleted

Indicate on the Circo plots in Figure 1 the deleted regions within the viral genome for each W12 subclone

Discuss the differences in the distribution of viral-host reads across the HPV genomes for the different W12 subclones and any effects on viral and host gene expression profiles. For example, why is the E1 gene devoid of reads in most subclones? Is that where the integration breakpoint occurred in these subclones? This point is later clarified in the discussion section – also include in the legend for Figure 1.

Lines 211-213 of the results section; the authors found that W12 clones F and A5 had the same integration site, with virus-host reads converging to the same region of chromosome 4. How do the transcriptional (viral and host) and epigenetic profiles of these clones compare?

Supplementary Figures 3-5C and 6D are difficult to interpret without having the matched DNA-seq data to visualize where the amplified regions are occurring within the host genome relative to the integrated viral sequences and this should be addressed in the figure. It is not clear what the red arrows are indicating in terms of how the viral genome was co-amplified along with the host sequence. The authors should combine data from Supplementary Figure 7 to better illustrate the architecture of each integration site

Lines 327-340; The authors show that the Hi-C heatmaps between clones D2 and G2 are similar across a 5 Mb window, suggesting that integration of the HPV16 genome does not affect the local genome architecture in these two subclones. The 5’ and 3’ integration breakpoints in both subclones are relatively close together (~25 Kb) and do not appear to disrupt any TAD boundaries. Furthermore, there are no complex rearrangements at these integration sites and so disruptions to local genome architecture might not be apparent at this 5 Mb scale. It is possible that integration sites that expand across a large genomic region (e.g. the 170 Kb deletion in clone H), display complex rearrangements and/or disrupt a TAD boundary would affect the local chromatin architecture. The authors should discuss this in the text and consider rewording the title of this results section as not to overstate this finding as this result was only confirmed in two subclones that have a relatively simple integration profile

Lines 346-347 and Figure 6; state which cell lines were used for alignment of integration breakpoints with TADs

Line 360; make it clear what is meant by “with no restriction to TADs”

Lines 365-369; The authors state “Compellingly, upon comparative analysis of Hi-C libraries between clones G2 and D2, a decrease in a host:host interaction was found within the TAD of integration, aligned with the HPV16:host interaction with the ARL15 gene. Hence, changes of gene expression within TADs are possibly due to modulated host:host interactions at this level.” Does this finding not contradict the conclusions presented in results section: HPV16 genome integration does not affect local host genome architecture (lines 326-340)?

Supplementary Figure 12A; the authors need to better describe in the figure legend what this panel is showing. It is not clear what is meant by “blue triangle” or what the comparison is between clone D2 and G2. Indicate the genomic position of the reference genome and the location of ARL15 and the integration locus. A colour scale should also be included.

Figure 8; The authors should provide further details of the distance from the integration site that genes are still differentially modulated in each subclone. For instance, what is the average size of the 5 gene bins and how does this relate to TADs/genomic architecture of the integration site?

Reviewer #3: (No Response)

PLOS authors have the option to publish the peer review history of their article (what does this mean?). If published, this will include your full peer review and any attached files.

Reviewer #1: No

Reviewer #2: No

Reviewer #3: No
---

## [Decision Letter · Decision Letter 1]

10 Jul 2021

Dear Groves,

Thank you very much for submitting your manuscript "Short- and long-range cis interactions between integrated HPV genomes and cellular chromatin dysregulate host gene expression in early cervical carcinogenesis" for consideration at PLOS Pathogens. As with all papers reviewed by the journal, your manuscript was reviewed by members of the editorial board and by several independent reviewers. The reviewers appreciated the attention to an important topic. Based on the reviews, we are likely to accept this manuscript for publication, providing that you modify the manuscript according to the review recommendations.

Each of the reviewers found the revised manuscript to be improved and to address most of the key points raised in the first round of review. While all of the points for the first two reviewers have now been met, there are still several issues highlighted by review #3 regarding bioinformatic analysis to address prior to publication. It will be important to more clearly state the analysis that suggests that recombination events have not caused association between human and viral DNA. It will also be important to state how the statistical analysis was chosen for the box plots in Figure 4, describing whether the distribution was normal and adding this to the legend, or performing the alternative tests that would be more appropriate if not, explaining whether the necessary grouping was done for the Fisher's exact test, add p-value corrections for multiplicity in Fig 8.

Sincerely,

Benjamin E Gewurz, M.D., Ph.D.

Associate Editor

PLOS Pathogens

Karl Münger

Section Editor

PLOS Pathogens

Kasturi Haldar

Editor-in-Chief

PLOS Pathogens

orcid.org/0000-0001-5065-158X

Michael Malim

Editor-in-Chief

PLOS Pathogens

orcid.org/0000-0002-7699-2064

Each of the reviewers found the revised manuscript to be improved and to address most of the key points raised in the first round of review. While all of the points for the first two reviewers have now been met, there are still several issues highlighted by review #3 regarding bioinformatic analysis to address prior to publication. It will be important to more clearly state the analysis that suggests that recombination events have not caused association between human and viral DNA. It will also be important to state how the statistical analysis was chosen for the box plots in Figure 4, describing whether the distribution was normal and adding this to the legend, or performing the alternative tests that would be more appropriate if not, explaining whether the necessary grouping was done for the Fisher's exact test, add p-value corrections for multiplicity in Fig 8.

Reviewer Comments (if any, and for reference):

Reviewer's Responses to Questions

**Part I - Summary**

Reviewer #1: The study by Graves et al. has addressed many of the concerns raised in the initial review. The authors conclusions have been tempered and confusing use of jargon has been clarified. The modified discussion of the significance of effects of integration on TADs and expression of associated genes is useful. The study provides some interesting insights into gene expression from integrated copies of HR-HPVs.

Reviewer #2: (No Response)

Reviewer #3: The strengths are good and weaknesses are addressed below under Minor Issues. The novelty, significance, execution and scholarship are good.

**Part II – Major Issues: Key Experiments Required for Acceptance**

Reviewer #1: None

Reviewer #2: (No Response)

Reviewer #3: None.

**Part III – Minor Issues: Editorial and Data Presentation Modifications**

Reviewer #1: None

Reviewer #2: (No Response)

Reviewer #3: The revised manuscript is much clearer.

The authors present striking and dramatic results in which HPV DNA that has integrated into the human genome can closely associate in cis with distally located human DNA and modulate gene expression at those locations. It’s proposed that the associated distal human DNA is not associated by recombination between the human DNA and HPV DNA, but by non-covalent interactions. Since covalent linkage is the most obvious explanation, the authors must ensure there are no recombination events that explain the association. The authors needed to show that the distal human DNA associated with HPV DNA was is not associated via covalent linkage by showing there are no aberrant junction reads or fragments from within the distal human DNA region that might explain its connection to HPV DNA. The authors state something to this effect in the response to reviewers but I don’t see anything in the manuscript so that the readers can know this. It should be clearly stated. Also, the authors should present the odds of not finding a junction should they exist near the distally associated human reads based on the number of reads in the area. This will give the readers confidence in the negative result.

Box plots in fig 4 are still confusing. If the distribution of values is not normal then non-parametric comparisons make sense, such as box plots shown, but is this true? If the data follows a normal distribution, then the means should be compared. What numbers are presented below the plots? The original manuscript said they were mean ± se but response to reviewers said median and legend now says nothing. If median then what are the ± values? Instead of a fairly standard test for comparing non-normal data, like Mann-Whitney, the authors use the Fisher’s exact test. While this reviewer can envision a way to group the values based on and relative to the median, it is not clear how the authors grouped the values for the Fisher’s exact test. This needs explaining.

The Fisher’s exact test was used for comparing the reads found in the region around the HPV integration site and another region, but it’s not clear how the authors did the necessary grouping.

For the F test for variance in bin gene expression in regions in Fig 8, p-values still need correction for multiplicity. Only four p-values in H, F and A5, look significant with this reviewer’s corrections using an FDR of 0.05. The FDR needs to be stated.

Line 391: “the mean of several clones lines” is not the comparator but a comparate; the means by which the comparates are compared is the comparator.

PLOS authors have the option to publish the peer review history of their article (what does this mean?). If published, this will include your full peer review and any attached files.

Reviewer #1: No

Reviewer #2: No

Reviewer #3: No

Figure Files:

Data Requirements:

Reproducibility:

References:

---

## [Editor Report · Decision Letter 2]

7 Aug 2021

Dear Groves,

We are pleased to inform you that your manuscript 'Short- and long-range cis interactions between integrated HPV genomes and cellular chromatin dysregulate host gene expression in early cervical carcinogenesis' has been provisionally accepted for publication in PLOS Pathogens.

Best regards,

Benjamin E Gewurz, M.D., Ph.D.

Associate Editor

PLOS Pathogens

Karl Münger

Section Editor

PLOS Pathogens

Kasturi Haldar

Editor-in-Chief

PLOS Pathogens

orcid.org/0000-0001-5065-158X

Michael Malim

Editor-in-Chief

PLOS Pathogens

orcid.org/0000-0002-7699-2064
---

## [Editor Report · Acceptance letter]

19 Aug 2021

Dear Groves,

We are delighted to inform you that your manuscript, "Short- and long-range cis interactions between integrated HPV genomes and cellular chromatin dysregulate host gene expression in early cervical carcinogenesis," has been formally accepted for publication in PLOS Pathogens.

Best regards,

Kasturi Haldar

Editor-in-Chief

PLOS Pathogens

orcid.org/0000-0001-5065-158X

Michael Malim

Editor-in-Chief

PLOS Pathogens

orcid.org/0000-0002-7699-2064